

# Lidar measurements of yawed wind turbine wakes: characterisation and validation of analytical models

Peter Brugger[1], Mithu Debnath[2], Andrew Scholbrock[2], Paul Fleming[2], Patrick Moriarty[2], Eric Simley[2], David Jager[2], Mark Murphy[2], Haohua Zong[1], and Fernando Porté-Agel[1]

[1]Wind Engineering and Renewable Energy Laboratory (WiRE), École Polytechnique Fedérale de Lausanne (EPFL), 1015 Lausanne, Switzerland
[2]National Renewable Energy Laboratory (NREL), 15013 Denver West Parkway, Golden, Colorado, 80401, USA

**Correspondence:** Peter Brugger (peter.brugger@epfl.ch)

**Abstract.** Wake measurements of a scanning Doppler lidar mounted on the nacelle of a yawed full-scale wind turbine are used for the characterization of the wake flow and the validation of analytical wake models. Inflow scanning Doppler lidars, a meteorological mast and the data of the wind turbine control system complemented the set-up. Results showed an increase of the wake deflection with the yaw angle that agreed with two of the three compared models. For yawed cases, the predicted power
of a waked downwind turbine estimated by the two previously successful models had an error of 17% and 24% compared to the SCADA data and 12% and 13% compared to the power estimated from the Doppler lidar measurements. Shortcomings of the method to compute the power coefficient in an inhomogeneous wind field are likely the reason for disagreement between estimates using the Doppler lidar data versus SCADA data. Further, it was found that some wake steering cases were detrimental to the power output due to errors of the inflow wind direction perceived by the yawed wind turbine and the wake steering
design implemented. Lastly, it was observed that the spanwise cross-section of the wake is strongly affected by wind veer, masking the kidney-shaped wake cross-sections observed from wind-tunnel experiments and numerical simulations.

## 1 Introduction

Wind turbines in wind farms can influence other turbines downstream and impact their performance. The interaction of the turbine rotor blades and the wind field creates a spatial volume of reduced wind speed and increased turbulence levels down-
stream of a wind turbine that can extend for several rotor diameters (Vermeer et al., 2003). This region is called the wake and affects downwind turbines negatively by decreasing power production and increasing fatigue loads (Thomsen and Sørensen, 1999). The spatial proximity of wind turbines in a wind farm and the wake effects on downwind turbines are an important source of power losses (Barthelmie et al., 2010).

Mitigating these wake effects on downwind turbines is an ongoing focus of research. Strategies that have been proposed
are adjusting the blade pitch angle and the generator torque (Bitar and Seiler, 2013), counter-rotating rows of wind turbines in wind farms (Vasel-Be-Hagh and Archer, 2017), optimizing the placement of the turbines within the wind farm based on terrain and wind climate (e.g. Shakoor et al., 2016; Kuo et al., 2016), or deflecting the wake away from the downwind turbine by introducing a yaw offset to the upwind turbine (Medici and Dahlberg, 2003). The latter approach, called wake steering or





active yaw control, is the focus of this paper. It utilizes the thrust force that the rotor imposes on the flow and by offsetting the rotor from the flow direction a transverse component of the thrust force is generated that displaces the wake from the line of the wind direction with the goal to deflect it away from the downwind turbine. While the power production of the yawed turbine is reduced, this loss is potentially overcompensated by the power gains of the downwind turbine (Bastankhah and Porté-Agel, 2015) and the strategy can be extended to a full wind farm (Gebraad et al., 2016).

Analytical models describe the wake of a yawed wind turbine based on a set of turbine and inflow parameters (Jiménez et al., 2009; Bastankhah and Porté-Agel, 2016; Qian and Ishihara, 2018). These models are computationally cheap compared to numerical simulations and therefore can be used to find a set of yaw angles that maximizes the power output (Gebraad et al., 2016; Fleming et al., 2019). Validation of the analytical models for yawed wind turbine wakes and studies on the effectiveness of the wake steering have been done with wind tunnel experiments (e.g. Bastankhah and Porté-Agel, 2016) and numerical simulations (e.g. Vollmer et al., 2016). However, studies of yawed wind turbines using field data are rare: Fleming et al. (2017a) and Annoni et al. (2018) analysed the wake deflection, the wake recovery, and the power output for an isolated yawed turbine; Fleming et al. (2017b) investigated the effects of wake steering on the power production for a yawed upwind and a waked downwind turbines at an offshore-site; and most recently Simley et al. (2020) investigated the influence of the wind direction variability on the achieved yaw offsets and power gains.

In this paper, field measurements, including inflow and wake measurements as well as wind turbine control system data from a wake steering upwind turbine and a waked downwind turbine, are used to (i) characterize the wake flow in terms of deflection, velocity deficit and width, (ii) to validate the wake deflection and power predicted from analytical models with the field measurements, and (iii) to analyse the performance of the wake-steering set-up implemented at this site.

## 2 Methods

This section introduces the measurement site, the instruments, the analytical models, and the data processing used to obtain the results. Indices are used to distinguish quantities measured by different instruments (see appendix 1 for an overview).

### 2.1 Research site and measurement setup

The measurement site is a large wind farm in northeast Colorado, United States. Measurements were conducted at an isolated cluster of five turbines at the north-western edge of the wind farm from 23 December 2018 until 06 May 2019 with the set-up shown in Fig. 1. The area north of the turbines is flat grassland and to the south and south-east is a downward terrain step of approx. 150 m followed by flat grassland. The measurement set-up consisted of an WindCube Doppler lidar and a meteorological mast that recorded vertical profiles of the wind speed and wind direction. A Stream Line Doppler lidar scanning the wake and a WindIris Doppler lidar scanning the inflow were installed on the nacelle of turbine 2 (T2). A thrid wake scanning Doppler lidar was installed on the nacelle of turbine (T3), but its data is not used in this study. Further, the SCADA data of T2 and T3 was provided by the wind park operator. This article focuses on conditions with northern wind directions with flat grassland upwind and no structures or turbines affecting the inflow.



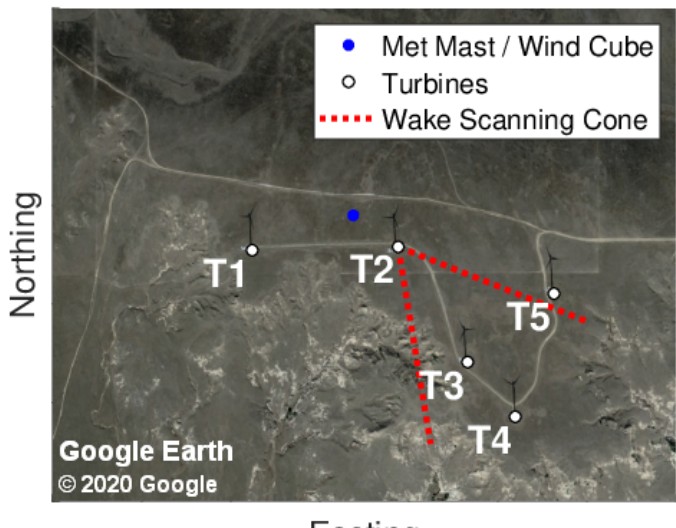

**Figure 1.** Overview of the measurement site and set-up. Shown in white are the five turbines of the local cluster with the remainder of the wind park to the east. Turbine 2 (T2) was programmed to introduce a yaw offset, if turbine 3 (T3) was downwind. The distance between T2 and T3 is approx. 390 m. T2 had two Doppler lidars installed on the nacelle to scan the inflow and the wake (Sect. 2.2.2 and 2.2.4). Shown in red is the scanning cone of the wake scanning Doppler lidar for a case with the wind direction aligned with the direction to T3 and a yaw angle of $20°$. Shown in blue is the location of the meteorological mast and the Wind Cube (Sect. 2.2.3 and 2.2.1).

The wind turbines were of the type 1.5SLE from General Electric Energy with active blade pitch control and a rated capacity of 1500 kW. Their hub height $z_{hub}$ is 80 m and the rotor diameter $D$ is 77 m. T2 was equipped with a yaw controller to introduce a wind speed dependent yaw offset for wind direction between $324°$ and $348°$ to deflect the wake from T3 (Fig. 2a). A negative yaw offset is a counterclockwise rotation of nacelle viewed from above. The power curve and pitch control of T2 are shown in Fig. 2b and 2d and for T3 in Fig. 2c and 2e. In absence of manufacturer information or measurement data for the thrust coefficient and due to the similarity of the thrust coefficient for most commercial wind turbines, the assumed thrust coefficient curve of the wind turbine follows the ensemble average shown in Fig. 2d. For a yawed turbine the thrust coefficient is adapted with $\widetilde{C}_T = C_T \cos^{1.5} \gamma$ (Bastankhah and Porté-Agel, 2017) and the power coefficient is modified with $\widetilde{C}_P = C_P \cos^3 \gamma$ (Adaramola and Krogstad, 2011), which includes the reduction of the rotor swept area. The readings of the nacelle position in the SCADA data of T2 were incremented by $4°$ on 17 January 2019 without affecting the true nacelle position to remove a bias between the the wind direction perceived by T2 and the WindCube. If the nacelle position of T2 is used to compute the position of T3 within the field of view of the wake scanning lidar this manipulation is reversed.

## 2.2 Measurement instruments

The instruments for the inflow and the wake measurements are introduced.



**Figure 2.** Characteristics of the wind turbines used for the wake steering test. The top panel (a) shows the target yaw offset as a function of the wind speed and wind direction for T2. Panels b) and d) show the power coefficient and the pitch angle of the blades as a function of the wind speed for T2 with the SCADA data shown as blue dots and the bin averages in red. Panels c) and e) show the same for T3. The bottom panel (f) shows in black the thrust coefficient curves of six wind turbines from manufacturer data (first compiled by Abdulrahman (2017)) and in red the ensemble average assumed as $C_T$ curve for T2.



### 2.2.1 WindCube

A WindCube Doppler lidar (version 2, manufactured by Leosphere and NRG Systems, Inc.) was located north of T2 and measured vertical profiles of the wind speed and the wind direction of the inflow (Fig.1). The lidar uses a laser wavelength of 1.54 $\mu$m and internally computes the wind speed ($U_{WC}$) and wind direction ($dir_{WC}$) from a PPI scan with an azimuth step 90° and an elevation angle of 62° followed by a vertical beam with the Doppler beam swinging technique assuming horizontal homogeneity (similar to the lidar in Lundquist et al. (2017)). The measurement data was filtered with a signal-to-noise ratio (SNR) threshold of −19 dB. The WindCube was set-up to provide the vertical profiles from 40 m a.g.l. to 260 m a.g.l. with a height resolution of 20 m and an averaging time of 1 minute. The WindCube data is available from 06 January 2019 until 09 March 2019. Further, the yaw angle ($\gamma_{WC}$) can be computed from the difference between the wind direction at hub height and the nacelle position of the T2.

### 2.2.2 WindIris

A WindIris Doppler lidar (manufactured by Avent Lidar Technology) was mounted on the nacelle of T2. The WindIris uses a 4 beam geometry with measurements at ±15° from the rotor axis in horizontal and ±5° in the vertical. The WindIris provides the wind direction relative to the rotor axis ($\gamma_{WI}$), the wind speed ($U_{WI}$) and the longitudinal turbulence intensity ($TI_{WI}$) for an upwind distance of 50 m to 200 m from the turbine and heights of 45 m a.g.l. to 125 m a.g.l. Its measurements are within the induction zone of the turbine and only vertically averaged measurements from a upwind distance of 90 m are used as a compromise between good data availability and a large upwind distance. The WindIris had problems that led to data loss during the campaign, which limits data availability to 12, 16, and 19 January and a long period from the 24 January 2019 until the 07 April 2019.

### 2.2.3 Meteorological Mast

A meteorological mast was located north-west of T2 next to the WindCube. The wind direction from the wind vanes at 38 m a.g.l. ($dir_{MM,38m}$) and 56 m a.g.l. ($dir_{MM,56m}$), the wind speed of the ultrasonic anemometer at 50 m a.g.l. ($U_{Sonic}$), and the wind speed of the cup anemometer at 60 m. a.g.l. ($U_{MM}$) will be used. The wind vanes had an alignment issue until the week of 11 February 2019, when they were replaced with freshly calibrated units and the cup anemometer had periods of suspicious measurements that might be connected to icing of the instrument. For those reasons, the wind measurements from meteorological mast are only used for validation of the WindCube. Further, the meteorological mast measured air temperature and air pressure near the surface from which the density of dry air $\rho_{MM}$ is computed.

### 2.2.4 Stream Line

A Stream Line Doppler lidar (manufactured by Halo Photonics Ltd.) was mounted on the nacelle of T2 scanning the wake downwind of the turbine. It performed an hourly scan schedule consisting of 2D and 3D scans of the wind field downwind. The 2D scans were horizontal swipes at an elevation of 0° covering an azimuth range from 160° to 220° with an azimuth step of

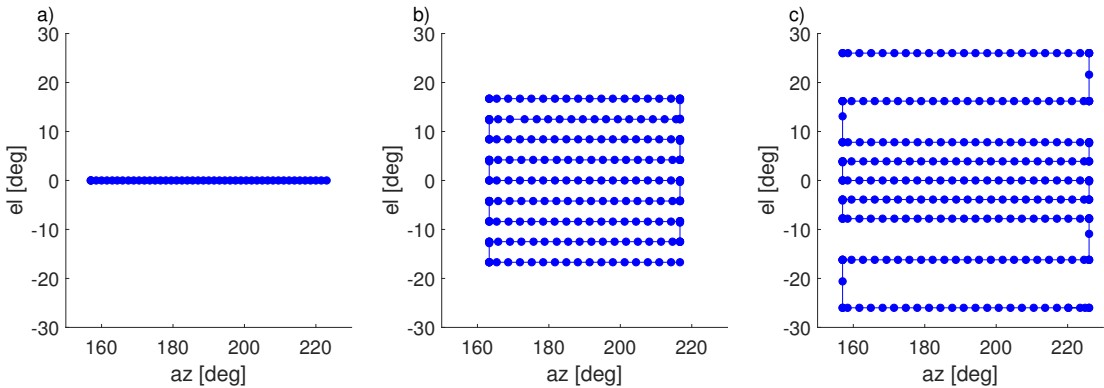

**Figure 3.** The scan pattern of the 2D (a) and the 3D scans with equal spaced elevation levels (b) and elevation levels with larger spacing at the top and bottom (c). The path of the scanner is shown as a blue line with measurement points indicated as blue points.

$1.5°$ (Fig. 3a), which were repeated 53 times back and forth within a 28-minute period. The 3D scans consisted of PPI swipes at 9 elevation angles, which were repeated between 20 and 22 times within a 31-minute period. The 3D scan pattern was iterated throughout the campaign with changes to the covered azimuth range and positions of the elevation levels (compare Fig. 3b and 3c). These changes were made to capture the wake at short downwind distances, but have little effect on the measurements
of the wake flow at the position of the downwind turbine. Further, other scan patterns were introduced to the scan schedule during the campaign, but those are not used in this study. The StreamLine system had an azimuth misalignment from the rotor axis of $-0.15°$ after installation on the nacelle. Levelling of the instrument is affected by tower movements, but their effects on the beam positions are mitigated by a grid-based post-processing of the measurement data introduced in the following section.

## 2.3   Data processing

The processing of the measurement data is introduced in the order in which it was done to obtain the results.

### 2.3.1   Inflow measurements and data selection

The 10-minute and 30-minute mean values and standard deviations of the wind speed, wind direction, and yaw angle were computed from the data of the WindCube, WindIris, meteorological mast and SCADA data. A filter was used to identify suitable intervals for further processing of the wake scanning lidar. The filter criteria are as follows:

– Data is available for the WindCube, the WindIris, and the SCADA data of T2 and T3.

  – Wind speed from the WindCube and WindIris is between $4\ \mathrm{m\ s^{-1}}$ and $15\ \mathrm{m\ s^{-1}}$.

  – Neither T2 nor T3 had a downtime and the rotor was turning.

  – The 10-minute period comprising a 30-minute period had changes of less than $3\ \mathrm{m\ s^{-1}}$ for the wind speed, and less than $5°$ for the wind direction.



Further, the 30-minute periods had to satisfy one of the two following conditions to be classified either as a wake steering case
or a control case:

–  Wake steering cases: north-western inflow with the WindCube wind direction between $320°$ and $350°$, active yaw control
of T2 (compare Fig. 2a), and the mean yaw angle between $3°$ and $30°$ for both WindIris and WindCube.

–  Control cases: north to north-eastern inflow with the WindCube wind direction between $0°$ and $75°$ and the yaw angle
between $-3°$ and $3°$ for both WindIris and WindCube.

The processing of the wake scanning Stream Line Doppler lidar described in the next section was carried out for periods that
satisfied the above filtering criteria. Periods were rejected at later stages if the measurements of the Stream Line system were
not available or the SNR filter rejected measurements in the investigated scan area. Because the selection of suitable periods
described here is based on 30-minute periods, but the 2D and the 3D scans of the Stream Line Doppler lidar were 28 and 31
minutes long, respectively, the final inflow parameters used for the results were re-computed for the precise scan durations at a
later stage.

### 2.3.2  Processing of wake scanning Doppler lidar data

For the suitable periods identified in the previous section, the data of the wake scanning Stream Line system was processed
along the following steps:

–  A signal-to-noise ratio (SNR) filter with a threshold of $-17$ dB was applied to remove low quality data points. If the
mean SNR at hub height was too low at a distance of $4D$, the scan was rejected altogether (e.g. periods with aerosol free
air or fog).

–  The azimuth angle of each lidar beam was adjusted so that the measurements were fixed in space relative to the ground
by removing changes of the nacelle position recorded in the SCADA data. The transformation is given by

$$az_{wsl,i} = az_{wsl,i} + (az_{SC,i} - \overline{az_{SC}}) \qquad (1)$$

with $az_{wsl,i}$ the azimuth angle of the $i$-th beam during the scan, $az_{SC,i}$ is the nacelle position of T2 at the time of the
measurement, and $\overline{az_{SC}}$ the angular mean nacelle position for the scan duration. A rejection of periods with excessive
nacelle position changes was not necessary, because the stationarity criterion of the wind direction in the previous section
already removed periods with large changes of the nacelle position.

–  The measurements were rotated into the mean wind direction such that it aligned with $az = 180°$ of the wake scanning
lidar with

$$az_{wsl,i} = az_{wsl,i} + \gamma. \qquad (2)$$

–  The radial velocity measured by the Doppler lidar was transformed to the longitudinal velocity based on elevation and
azimuth angles, sorted into a regular spherical coordinate system, and interpolated on a Cartesian coordinate system with





10 m resolution. These procedures are described in Fuertes et al. (2018) for the 2D scans and in Brugger et al. (2019) for the 3D scans.

The above steps provided the longitudinal mean velocity field $u_{2D}(x,y)$ and $u_{3D}(x,y,z)$ in a Cartesian right-hand system with origin at the nacelle of T2 and the $x$-axis pointing in wind direction and the $z$-axis upwards. The corresponding velocity deficits are then given by

$$\Delta u_{2D}(x,y) = u_{WC}(80 \text{ m}) - u_{2D}(x,y) \tag{3}$$

and

$$\Delta u_{3D}(x,y,z) = u_{WC}(z) - u_{3D}(x,y,z) \tag{4}$$

with $u_{WC}(z)$ interpolated to the grid heights.

### 2.3.3 Wake center deflection

The wake was characterized by fitting a Gaussian function given by

$$g(\delta,\sigma,C) = C\exp\left(\frac{(y-\delta)^2}{\sigma^2}\right) \tag{5}$$

to $\Delta\overline{u}_{2D}(x,y)$ and $\Delta\overline{u}_{3D}(x,y,z_{hub})$ at each downwind distance. The fit used a Gaussian weighting function with a width of $1.5\sigma$. The position of the peak given by $\delta(x)$ is equivalent to the wake deflection, because the coordinate system was rotated into the wind direction (Eq. 2). To remove cases where the Gaussian fit was influenced by the wakes or the hard targets of neighbouring turbines and to ensure that only results within the far wake are used, the result was rejected if the correlation coefficient of the Gaussian fit and the measurement data was below $0.99$ at $x/D = 4$ (a visual verification showed that all instances of this problem were detected).

### 2.3.4 Power

The power of the upwind turbine (T2) was computed from the inflow measurements of the WindCube with the assumption that the inflow is horizontally homogeneous across the rotor area. It is then given by

$$P_{WC} = \frac{1}{2}\rho_{mm}C_{P,T2}\cos^3\gamma \iint\limits_A u_{WC}^3(z)\,\mathrm{d}y\mathrm{d}z, \tag{6}$$

with the rotor area $A$ defined by $\sqrt{y^2 + (z-z_{hub})^2} \leq 0.5D$ and $C_{P,T2}$ was interpolated from the power curve of T2 shown in Fig. 2 based on the $U_{WC}(z_{hub})$. For the downwind turbine (T3), the power was computed from the longitudinal velocity field of the wake scanning lidar by integration over the rotor area. It is given by

$$P_{wsl} = \frac{1}{2}\rho_{mm}C_{P,T3} \iint\limits_A u_{3D}^3(4D,y,z)\,\mathrm{d}y\mathrm{d}z, \tag{7}$$



with $\sqrt{(y - y_{T3})^2 + (z - z_{hub})^2} \leq 0.5D$ and $y_{T3}$ the transverse position of the T3 in the coordinate system aligned with the wind direction. The power coefficient was interpolated from the power curve of T3 based on the average velocity across the rotor area for T3. The integrals were approximated by sums according to the grid resolution of the measurement data.

### 2.4 Analytical Models

Three analytical models are compared with the field measurements in this study. The analytical models were introduced by Jiménez et al. (2009), Bastankhah and Porté-Agel (2016), and Qian and Ishihara (2018), respectively, and their equations are presented in Appendix 2. All three models use the longitudinal turbulence intensity, the yaw angle, and the thrust coefficient as input variables and predict the longitudinal velocity deficit field $\Delta u_{mod}(x, y, z)$ of the wake. The models are computed for the same 10 m resolution Cartesian coordinate system as the velocity fields of the wake scanning lidar for consistency. Together with the inflow measurements of the WindCube, the longitudinal velocity field is computed with

$$u_{mod}(x, y, z) = \Delta u_{mod}(x, y, z) + u_{WC}(z), \tag{8}$$

where $u_{WC}(z)$ is interpolated to the grid levels. The turbine power of T3 is then computed from the model analogous to Eq. (7), but with the predicted longitudinal velocity field of the analytical model instead of the velocity field from the lidar measurements.

## 3 Results and Discussion

The time frame of analysis is 6 January 2019 until 9 April 2019, because outside of that time frame data of the WindIris or WindCube was missing. The synoptic conditions were characterized by the winter season with daily mean temperatures mostly between $-10\,^{\circ}\text{C}$ and $5\,^{\circ}\text{C}$. The main wind directions were north-west and south-east with wind speeds up to $25\,\text{m s}^{-1}$ (Fig. 4). Periods of clear air and snow or fog events further reduced data availability of the remote sensing instruments.

### 3.1 Inflow

The inflow measurements, especially of the yaw angle, are essential for the quality of the results presented in the following sections. Therefore, an inter-comparison of the inflow measurements for wind speed, wind direction, and yaw angle will be presented first.

The wind speed from the WindCube, ultrasonic anemometer, cup anemometer and WindIris are compared in Fig. 5. All available data during the analysis time frame is used for the comparison irrespective of the filtering criteria (Sect. 2.3.1). The WindCube shows good agreement to the ultrasonic anemometer with correlation coefficient of 0.99, a slope near unity and a RMSE of $0.68\,\text{m s}^{-1}$ (Fig. 5a). The underestimation at high wind speed might be explained by the height difference. The agreement between the WindCube and the cup anemometer is also good with correlation coefficient of 0.98, a slope near unity and a RMSE of $1.00\,\text{m s}^{-1}$ (Fig. 5b). However, here we removed two periods of data around the 07 February 2019 and 26 February 2019 were the cup anemometer showed consistently low wind speeds. Both periods coincide with very low



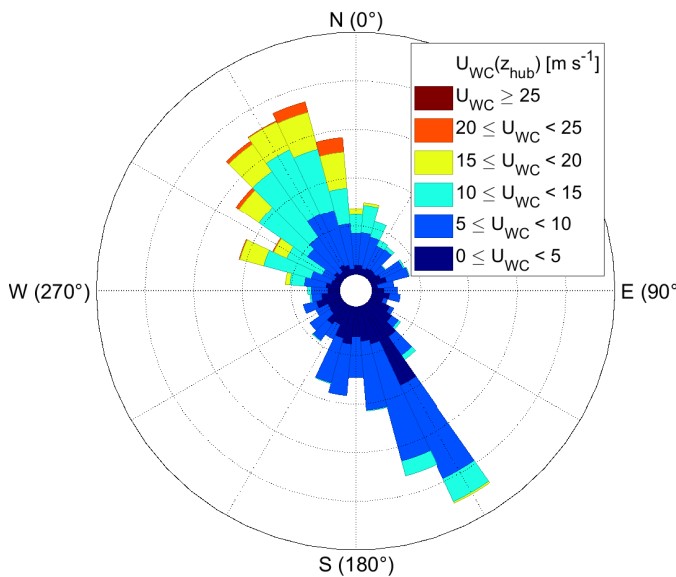

**Figure 4.** Wind rose based on the $u_{WC}$ and $dir_{WC}$ at hub height using data from 6 January 2019 until 09 April 2019. Software written by Daniel Pereira was used to create the wind rose (https://www.mathworks.com/matlabcentral/fileexchange/47248-wind-rose, MATLAB Central File Exchange, Retrieved 11 December 2019).

temperatures according to the air temperature measurement at 2 m and icing of the cup anemometer might have played a role here. An underestimation at high wind speeds is not observed. The WindCube and the WindIris show systematic deviations due to the induction zone of the wind turbine (Fig. 5c). Based on this comparison, the wind speed of the WindCube will be used in the following, because it is available at hub height, not influenced by the induction zone, and compares well with the
210 ultrasonic and cup anemometer.

The wind direction from the WindCube and the two wind vanes of the meteorological mast have an offset to each other until the 11 February 2019 after which the mast data is unavailable for five days due to maintenance. After the 16 February 2019, both heights agree with the WindCube (Fig. 6). As for the wind speed above, the filtering criteria are not applied here. The RMSE is $5.54°$ for the lower wind vane and $6.25°$ for the upper wind vane. The wind direction of the WindCube and the
215 wind vane has a correlation coefficient of 1.00 and a slope near unity at both heights. As for the wind speed, the WindCube will be used as reference for the wind direction, because it agrees well with meteorological mast after its maintenance, so it is presumably also correct before.

The yaw angle from the WindIris, the SCADA data, and the WindCube are compared (Fig. 7). The data filtering criteria of Sect. 2.3.1 were applied here, but without the yaw angle restriction for the control cases. This limitation to northern inflow
220 directions for the comparison is due to the wakes of the neighbouring wind turbines affecting the comparison negatively for other inflow directions. For the non-yawed control cases, a RMSE of $1.82°$ was found between the SCADA data and the



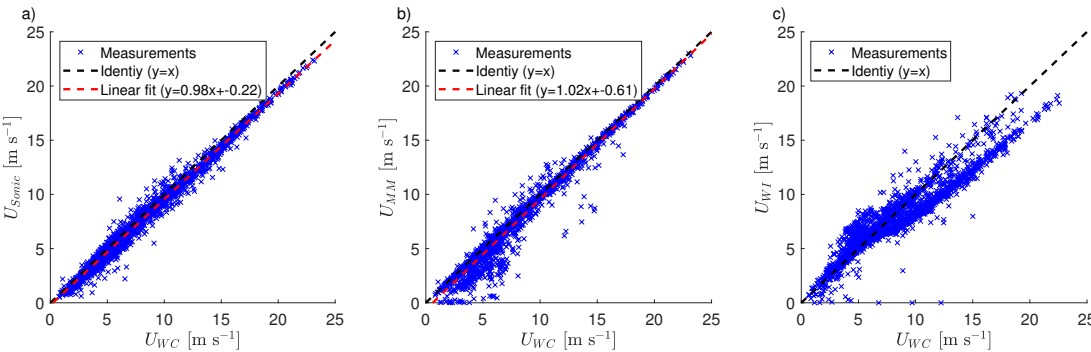

**Figure 5.** Inter-comparison of the inflow wind speed measurements between the ultrasonic anemometer at 50 m and the WindCube at 60 m (left panel, a), the meteorological mast at 60 m and the WindCube at 60 m (middle panel, b), and the WindIris and the WindCube at hub height (right panel, c). The black dashed line shows the identity $x = y$ and a linear fit is shown as a red dashed line.

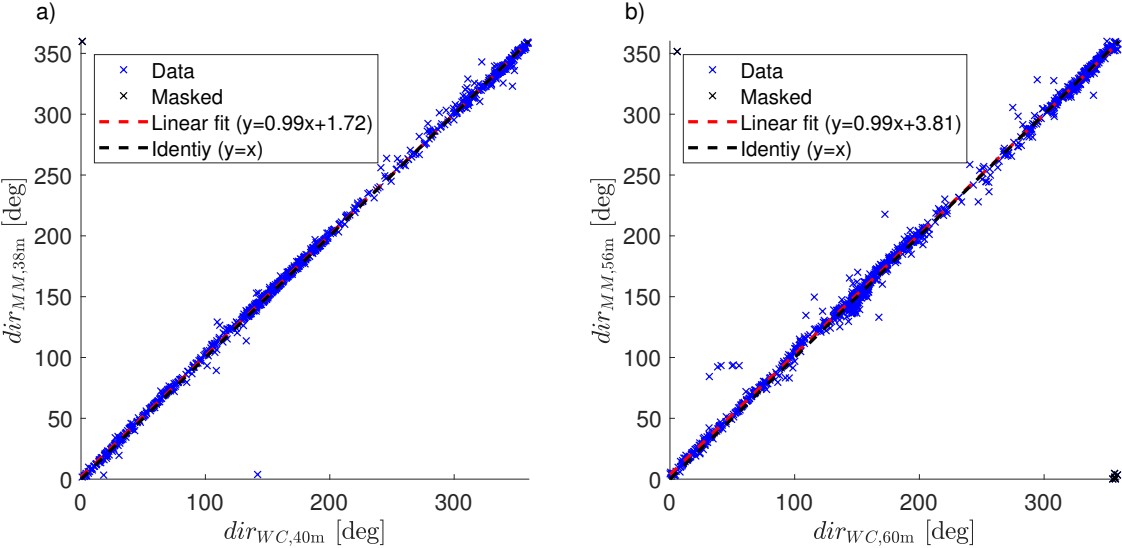

**Figure 6.** Comparison of the inflow wind direction measurements from the WindCube and the meteorological mast for 40 m (left panel, a) and 60 m (right panel, b) after 16 February 2019. The black dashed line shows the identity $x = y$ and a linear fit is shown as a red dashed line. Black crosses are data points that were excluded from the linear fit.





WindIris. The Gaussian fit suggests a bias of $1°$ between the instruments (Fig. 7a). The WindCube and the WindIris have a RMSE of $1.30°$ and a smaller bias than the SCADA data for the control cases (Fig. 7c). For the wake steering cases, a large bias between the WindIris and the SCADA data can be seen for $\gamma < -5°$ (Fig. 7b), that is not present between the WindCube and the WindIris (Fig. 7d). That observation suggests that yawing of the wind turbine affects the measurements of the wind vane on top of the nacelle. This view is supported by an increase of the RMSE to $2.10°$ between the WindIris and the SCADA data, while the RMSE between the WindCube and the WindIris did only change slightly to $1.32°$ for the wake steering cases.

### 3.2 Wake deflection

The deflection of the wake center from the downwind direction is investigated. The wake deflection is evaluated at a downwind distance of $x/D = 4$ to avoid the influence of hard targets or wakes of neighbouring turbines at larger distances and the near wake at smaller distances (Fig. 8).

First, the wake deflection is verified for non-yawed control cases, where no wake deflection is expected. The distribution of the normalized wake deflection using the WindIris has a RMSE of $0.08$ (Fig. 9a) and using the wind direction of the WindCube with the nacelle position of T2 provides a RMSE of $0.07$ (Fig. 9b). These errors agree with the RMSE of the yaw angle between the two instruments ($4\sin(1.30°) = 0.09$) and both distributions have mean value that is not significantly different from zero. The consistency between the yaw angle errors and wake deflection distribution shows that the wake scanning and its spatial positioning were working well, and the absence of a bias shows that the alignment of the wake scanning lidar with the rotor axis is correct (the measured offset of $0.15°$ during the installation was taken into account in the processing). Since we could not identify a clear favourite between the WindIris and the WindCube for the yaw angle, the average of both will be used for the remainder of the article.

The wake deflection for the wake steering cases is shown in Fig. 9c. The observed wake deflection increases with the yaw angle as expected from wind tunnel experiments (Bastankhah and Porté-Agel, 2016) and numerical simulations (Lin and Porté-Agel, 2019). The analytical model of Jiménez et al. (2009) overestimates the wake deflection and the models by Bastankhah and Porté-Agel (2016) and Qian and Ishihara (2018) better match the wake deflection from the field measurements. The overestimation of the Jiménez et al. (2009) model was also observed by Bastankhah and Porté-Agel (2016) with wind tunnel experiments and by Lin and Porté-Agel (2019) with numerical simulations. The measurement data shows considerably larger scattering than the model predictions, which is likely a consequence of the remaining non-stationarity of the atmospheric boundary layer in the dataset and the errors of the measurement data. It should be noted that the short downwind distance of $x/D = 4$ at which the models are evaluated is heavily influenced by the wake skew angle assumed for the near wake, which is used to provide an initial condition for the far-wake. The similar wake deflections for Bastankhah and Porté-Agel (2016) model and Qian and Ishihara (2018) model are then explained by the identical wake skew angle used by both models (Eq. 10 and Eq. 22) and noticeable differences of the wake deflection between these two models only appear at larger $x/D$ (Lin and Porté-Agel, 2019). Further, for cases with very low turbulence intensities, the models were not able to make a prediction at $x/D = 4$, because the predicted length of the near wake was longer. However, since the turbine spacing within a wind farm can be relatively short (here $5D$), this highlights the importance of a near wake description in the analytical models.

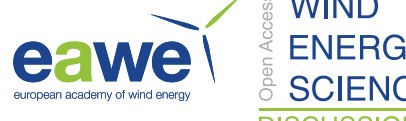

**Figure 7.** Inter-comparison of the yaw angle measurements. The top left panel (a) shows a histogram of the yaw angle difference between the WindIris and the SCADA data of T2 for the control cases. The top right panel (b) shows the yaw angle from the SCADA data of T2 and the WindIris for wake steering cases. The bottom left panel (c) shows a histogram of the yaw angle difference between the WindCube and the WindIris for the control cases. The bottom right panel (d) shows the yaw angle from the SCADA data of T2 and the WindIris for wake steering cases. The red line shows a Gaussian fit to the histogram and the black dashed line is the identity. The data was filtered according to Sect. 2.3.1, but for panel (a) and (c) the yaw angle limitation was omitted.



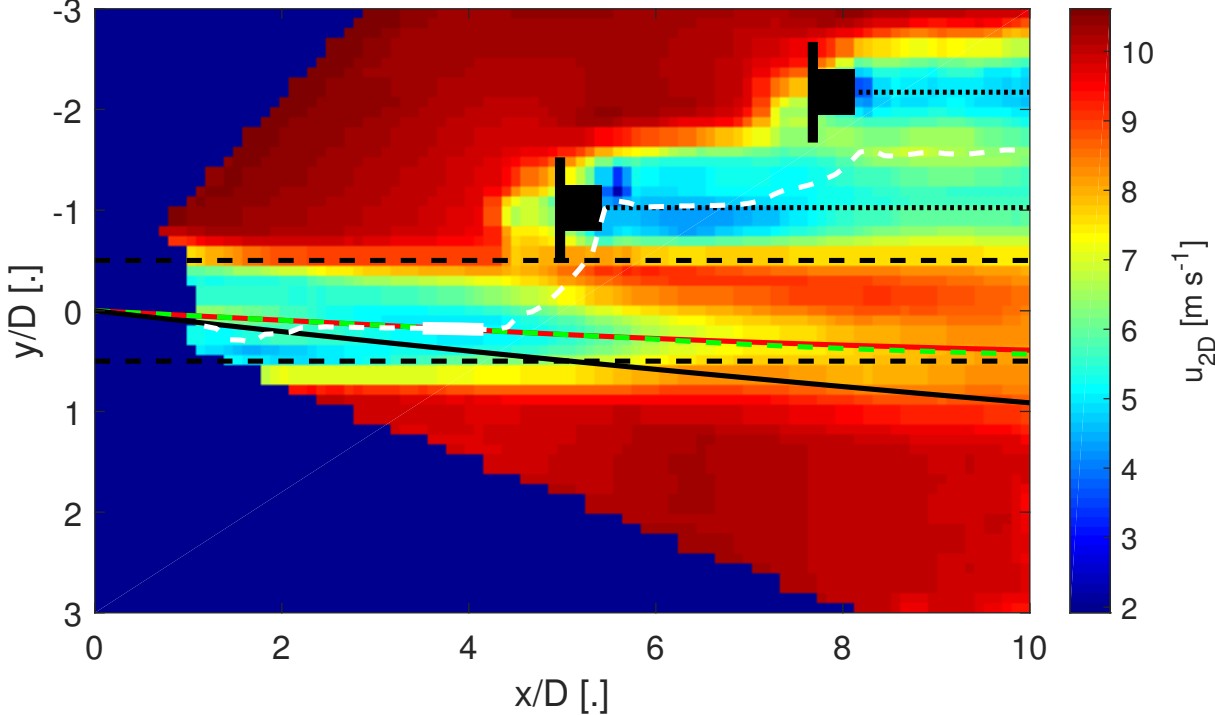

**Figure 8.** Example case of a yawed wind turbine wake with mean yaw angle of $\gamma = 18°$. The mean longitudinal velocity field is shown as a colour image. The predicted wake deflection of the Bastankhah and Porté-Agel (2016) model as a red solid line, the Qian and Ishihara (2018) model as a green dashed line, and the Jiménez et al. (2009) model as a black solid line. The dashed white line shows the result of the wake center detection and the solid white line indicates the part with a correlation coefficient larger than 0.99 (see Sect. 2.3.3). The black dashed line indicates the rotor area of T2. Turbine 3 and 4 are stylized in black and a black dotted line as an visual aid to indicate the downwind direction.

## 3.3 Power

The power estimated from the velocity field predicted by the analytical models is investigated. First, the predictions of the three analytical models are validated against the SCADA data and the measurements of the wake scanning lidar. Then, the effect of wake steering on the power of the downwind turbine (T3) and the full system of upwind and downwind turbine (T2+T3) is 260 investigated. The investigation is carried out for periods classified as wake steering cases with a 3D scan of the wake scanning lidar.

### 3.3.1 Model validation for power

The power estimated from the measurements of the Doppler lidars (Eq. 6 and Eq. 7) is compared with the SCADA data (Fig. 10). The power of T2 computed from the WindCube measurements and the SCADA have correlation coefficient of 0.98

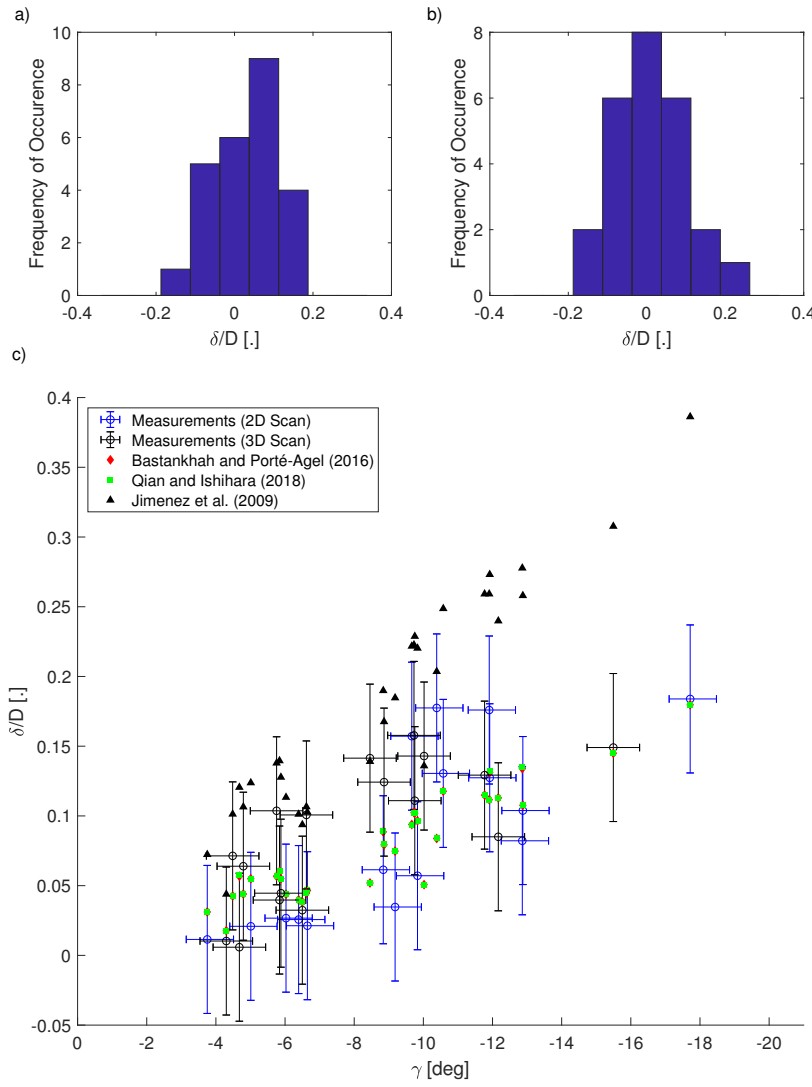

**Figure 9.** The two top panels show a histogram of the normalized wake deflection $\delta/D$ at $x/D = 4$ for the control cases based on the yaw angle $\gamma$ from the WindIris (a) and the WindCube wind direction and the nacelle position of T2 (b). The bottom panel (c) shows the normalized wake deflection at $x/D = 4$ as a function of the yaw angle for the wake steering cases. Here, the yaw angle from WindIris and WindCube were averaged. The measurements are shown in blue for the 2D scans and in black for the 3D scans. The errorbars are based on the errors found between WindIris and WindCube (Sect. 3.1). The analytical models of Jiménez et al. (2009) (black triangles, Eq. A25), Bastankhah and Porté-Agel (2016) (red diamonds, Eq. A7), and Qian and Ishihara (2018) (green squares, Eq. A18) are plotted for each case.





and a RMSE of 69 kW (Fig. 10a). The power differences between the WindCube and the SCADA data show no relationship to the yaw angle indicating that the adjustment of the power coefficient of a yawed turbine with $\cos^3 \gamma$ as proposed by Adaramola and Krogstad (2011) holds for the field data. The power of T3 has a correlation coefficient of 0.97 and a RMSE of 132 kW between the wake scanning lidar and the SCADA data (Fig. 10b). A bias is not apparent for T2 nor for T3. One possible reasons for the larger errors for T3 could be that the power coefficient curve used to compute the power is not ideal for cases with an

inhomogeneous wind field across the rotor, if T3 is partially waked as it is frequently the case in the data set. A second reason could be the influence of the induction zone in combination with a power curve that is based on the free stream velocity.

A comparison of the power computed from the Bastankhah and Porté-Agel (2016) model, the Qian and Ishihara (2018) model, and the Jiménez et al. (2009) model with the measurements is shown in Fig 11. The Qian and Ishihara (2018) model has a RMSE of 98 kW with the wake scanning lidar and 172 kW with the SCADA data with correlation coefficients of 0.98

and 0.93 respectively. The Bastankhah and Porté-Agel (2016) model has a RMSE of 103 kW with the wake scanning lidar and 193 kW with the SCADA data with the same correlation coefficients as the Qian and Ishihara (2018) model. The Jiménez et al. (2009) model has an RMSE of 211 kW with the wake scanning lidar, 224 kW with the SCADA data and correlation coefficients of 0.96 and 0.92, respectively. The Jiménez et al. (2009) model has considerably larger errors than the other two models, because it assumes a top-hat velocity deficit that overestimated the velocity deficit at the edges of the wake, which

resulted in an underestimation of the power for a partially waked downwind turbine. The Gaussian velocity deficits of the other two models better matched the Doppler lidar observations in this respect. The better agreement with the wake scanning lidar than with the SCADA data supports the assumption that the power coefficient has problems with partially waked turbines. Several factors contribute to the error of the analytical models: the physical simplifications, the errors of the input parameters, and the errors from the inflow scanning WindCube that propagate into the longitudinal velocity field. Assuming the error

between the power estimated from the WindCube and the SCADA data of T2 as a proxy for the propagated error, the Qian and Ishihara (2018) model and the Bastankhah and Porté-Agel (2016) model would have a RMSE with the SCADA that is comparable to the wake scanning lidar. Using different methods to estimate the power coefficient does not affect the overall findings (e.g. using the velocity in front of the nacelle instead of averaging the rotor area or switching between the model prediction and the lidar measurement).

### 3.3.2 Effect of wake steering on the power: example case

The dataset is searched for pairs of 30-minute periods with T3 downwind of T2 and similar inflow conditions, but one being yawed and the other not. All periods where the wind direction was aligned with the downwind turbine within $1°$ were ordered by the wind speed and two suitable pairs were identified (Fig. 12a and 12b). In case of the second pair, the turbulence intensity was too low for the analytical model to make a prediction at $x/D = 4$ and therefore only the first pair is discussed in the

following.

The inflow measurements and the power output of the turbines of the example case are summarized in Table 1a and the longitudinal mean velocity fields of the wake scanning lidar are shown in Fig. 12d and 12e. The increase of wind speed and the decrease of wind shear from the yawed to the non-yawed case together with the power losses of the yawed turbine could

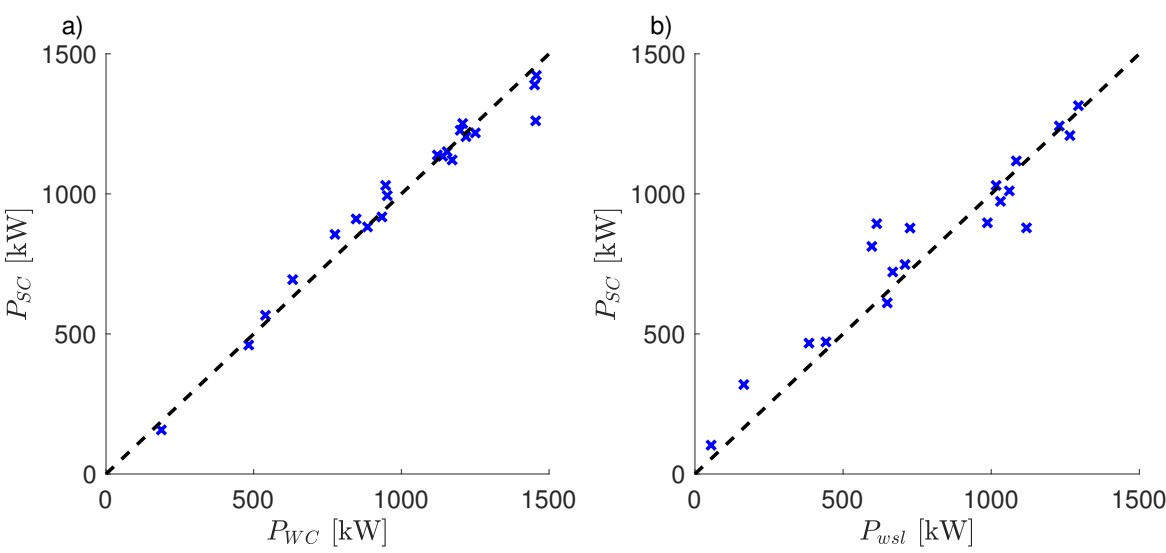

**Figure 10.** Comparison of the power from the SCADA data and the power estimated from the Doppler lidar measurements for T2 (a) and T3 (b). Blue crosses show the measurement data and the black dashed line is the identity ($y = x$).

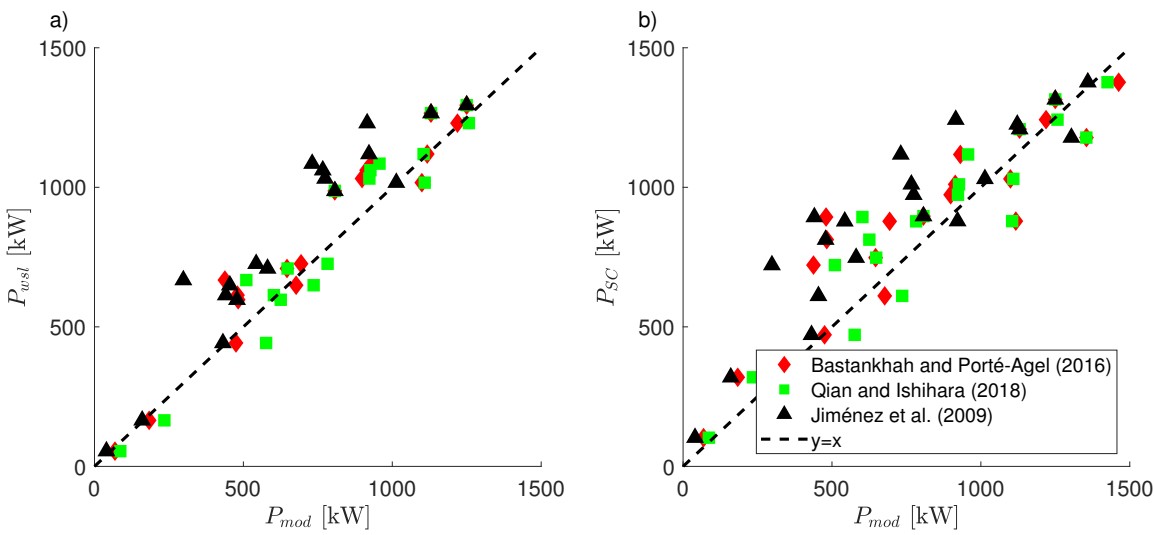

**Figure 11.** The power prediction of the analytical models for T3 compared with the wake scanning lidar (a) and the SCADA data (b). Red diamonds show the Bastankhah and Porté-Agel (2016) model, green squares the Qian and Ishihara (2018) model, and the black dashed line is the identity.





**Table 1.** Inflow and power output for the yawed case (left column) and non-yawed case (right column) shown in Fig. 12d and 12e. The upper part (a) presents the inflow measurement from the Doppler lidars and power from the SCADA data. The lower three parts show the power estimated from the inflow profiling lidar for T2 and the prediction of the Qian and Ishihara (2018) model for T3 based on the inflow values (b), the averaged inflow values only varying $\gamma$ (c), and the inflow values with $\gamma = 0$ (d).

|  | Description |  | Yawed | Non-yawed |
|---|---|---|---|---|
| a) | Inflow and | $\gamma$ [deg] | -12.5 | -0.2 |
|  | SCADA | $dir_{WC}(z_{hub})$ [m s$^{-1}$] | 323.3 | 232.2 |
|  |  | $u_{WC}(z_{hub})$ [deg] | 10.3 | 10.5 |
|  |  | $TI_{WI}$ [.] | 0.05 | 0.07 |
|  |  | $P_{T2,SC}$ [kW] | 1134 | 1197 |
|  |  | $P_{T3,SC}$ [kW] | 894 | 790 |
| b) | Inflow and | $P_{T2,WC}$ [kW] | 1105 | 1183 |
|  | wake steer. | $P_{T3,mod}$ [kW] | 822 | 668 |
| c) | Averaged | $P_{T2,WC,avg}$ [kW] | 1093 | 1175 |
|  | inflow | $P_{T3,mod,avg}$ [kW] | 827 | 655 |
| d) | No wake | $P_{T2,WC,\gamma=0}$ [kW] | 1187 | 1183 |
|  | steering | $P_{T3,mod,\gamma=0}$ [kW] | 733 | 667 |

explain the power difference for T2 seen in the SCADA data. For T3, the SCADA data reports higher power for the case with wake steering compared to case without wake steering, which could be explained by the deflection of the wake.

Using the Qian and Ishihara (2018) model and the inflow measurements to predict the power of the turbines captures the tendencies, but underestimates the power for T3 (Table 1b). The effect of the wake steering can be isolated by averaging $TI_{WI}$ and $u_{WC}(z)$ for both cases and only varying $\gamma$ (Table 1c). Conversely, the effect of the inflow conditions can be isolated by setting $\gamma = 0°$ and using $TI_{WI}$ and $u_{WC}(z)$ as measured (Table 1d). The results confirm that the wake steering had an effect on the power of T3 and changes of the inflow alone cannot explain the power differences between the yawed and the non-yawed case. Based on the analytical model and the SCADA data, the yawed T2 lost 60–80 kW and T3 gained 90–170 kW by the wake steering. The Bastankhah and Porté-Agel (2016) model had qualitatively similar results, but approximately 20 to 30 kW smaller than the Qian and Ishihara (2018) model. As a side note, it was observed that wake steering is not necessary at high wind speeds, because the wake has enough available power for the downwind turbine to run at its rated capacity (Fig. 12c).

Using yawed and non-yawed cases with similar inflow conditions as above to investigate the effect of wake steering for a wider part of the data set is not feasible due to the limited number of suitable pairs. However, this example case illustrated that using an analytical model to artificially remove the wake steering captures the power changes and can be used to investigate the effect of wake steering on the power.





**Figure 12.** The inflow wind speed (a), the yaw angle of T2 (b), and the power (c) for all 30-minute periods with the wind direction aligned with the downwind turbine within 1° sorted by wind speed (data filtering of Sect. 2.3.1 not applied). Highlighted with circles are the two pairs with similar wind speed and wind direction and all measurement data available, but different yaw angles. The two bottom panels show the mean longitudinal velocity fields at hub height from the wake scanning Doppler lidar for the first pair with the non-yawed case on the left (d) and the yawed case on the right (e). The rotor area shadow of T2 is indicated as a black dashed line and the position of T3 is stylized in black.





### 3.3.3 Effect of wake steering on the power: complete data set

The effect of wake steering on the power is investigated using the periods classified as wake steering cases. The 12 and 24 January 2019 have been excluded from this part of the analysis, because the yaw controller had toggling issues. The data set will be divided into two groups based on the wind direction following a visual inspection of the volumetric lidar measurements, which showed two categories of wake steering cases:

1. Successful wake steering, where the wake of the yawed T2 was partially or completely deflected away from T3 (Fig. 13a
and 13b).

2. Unnecessary or harmful wake steering, where the wake of the yawed T2 would have missed T3 even if T2 would not have yawed (Fig. 13c and 13d) or where the wake of the yawed T2 was deflected on T3 instead of away (Fig. 13e and 13f).

The latter group is expected to be detrimental to the overall power output, because the unnecessary wake steering decreases the power output of the upwind turbine without gains for the downwind turbine and the harmful wake steering case decreases the
power output of both turbines. Geometrical considerations of the rotor area shadow of T2 in the wind direction can explain the unnecessary cases. The harmful wake steering cases were observed for wind directions very close or smaller than the direction towards T3 and might be explained by errors of the wind direction perceived by the wind turbine (Fig. 7b) or the variability of the wind direction during the scan period (Simley et al., 2019). Therefore, the wake steering cases are separated into two groups: a narrow inflow sector from $325°$ to $335°$ and a wide inflow sector from $310°$ to $350°$.

The effect of wake steering on the power is investigated based on the differences between the power predicted by the Qian and Ishihara (2018) model for the inflow parameters and a hypothetically non-yawed case with the same inflow conditions otherwise. The results for the downwind turbine are shown in Fig. 14 and summarized in Table 2. An average power improvement of 72 kW (or 9%) for T3 for the narrow group that is reduced to 22 kW (or 2%) for the wide group indicates that several periods in the wide group had small power gains or even power losses for T3. The combined system that includes the power
losses of the yawed T2 has a power improvement 23 kW (or 2%) for the narrow group and shows virtually no improvement for the wide group (2 kW or 0%). The Bastankhah and Porté-Agel (2016) model provided qualitatively similar results to the Qian and Ishihara (2018) model (T3 power gains of 11% for the narrow group that are reduced to 3% for the wide group). For both models, the harmful or unnecessary wake steering cases were reducing the power significantly for the wake steering set-up in this study. These findings are in line with Simley et al. (2020).

## 3.4 Shape of the wake

The kidney-shaped spanwise cross-sections of yawed-turbine wakes observed in wind tunnel experiments (Bastankhah and Porté-Agel, 2016) and numerical simulations (Howland et al., 2016; Lin and Porté-Agel, 2019) were not observed in the data from the field measurements. Using a point vortex transportation model introduced by Zong and Porté-Agel (2020) it is shown that the effect of wind veer, which is usually present in the atmospheric boundary layer, masked any effects of the yaw angle
on the shape of the wake (Fig. 15). This is in line with a simple assessment of the wake displacement based on the transversal







**Figure 13.** Examples for observed categories of wake steering. The top row (a,b) shows a successful wake steering case. The middle row (c,d), shows an unnecessary wake steering case. The bottom row (e,f) shows a harmful wake steering case. The colour scale shows the longitudinal velocity of the wake scanning Doppler lidar. The left column shows a horizontal cross-section of the longitudinal velocity at hub height. The right column shows a spanwise cross-sections of the longitudinal velocity at a downwind distance of $4D$. The red dashed lines and red solid circles show the outline of the rotor area of T2 in wind direction. The position of T3 is stylized in black and the black solid circle shows the rotor area of T3.





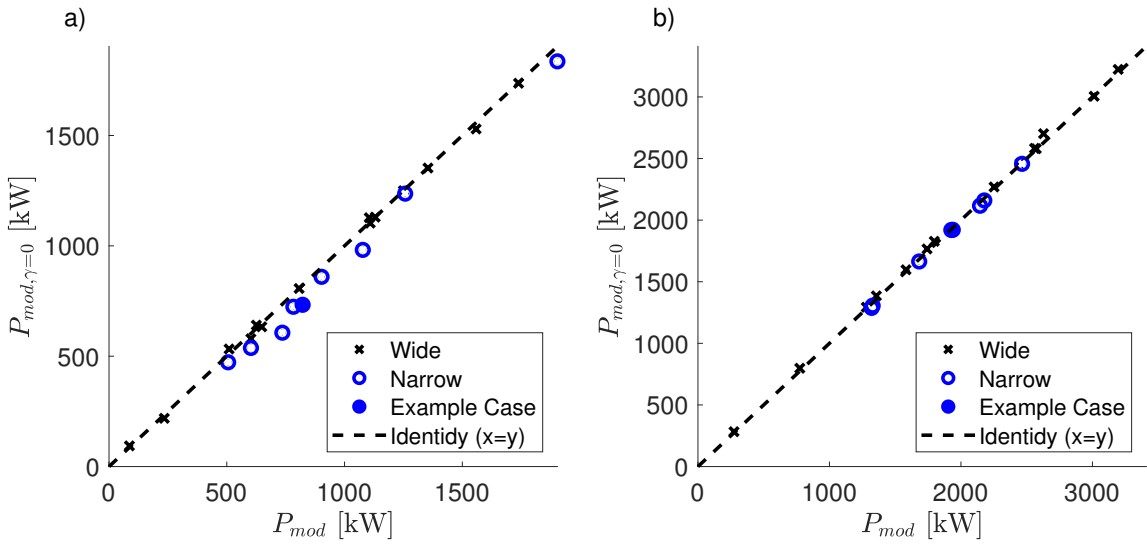

**Figure 14.** The effect of wake steering on the power based on the Qian and Ishihara (2018) model. The left panel (a) shows the effect on the power for the downwind turbine (T3) and the right panel (b) for the combined system of upwind and downwind turbine (T2+T3). The hollow blue circles indicate data points from the narrow inflow sector, the black crosses data points from the wide inflow sector, and solid blue circle is the yawed example case from Sect. 3.3.3.

**Table 2.** Maximum and average power gains and losses due to wake steering for the narrow and the wide group based on the Qian and Ishihara (2018) model. The two left columns show the downwind turbine (T3) and the two right columns show the combined system of upwind and downwind turbine (T2+T3). The percentage values are based on the power of the yawed case.

|  | T3 | | T2 + T3 | |
| --- | --- | --- | --- | --- |
|  | Narrow | Wide | Narrow | Wide |
| Max. gain | 18% | 18% | 3% | 3% |
| Avg. gain | 8% | 4% | 2% | 1% |
| Avg. loss | Non | -4% | 0% | -1% |
| Max. loss | Non | -8% | 0% | -3% |
| Overall | 8% | 3% | 2% | 0% |





advection due to the wind veer with

$$\Delta y = x \tan\left(\frac{\alpha_{tt} - \alpha_{bt}}{D}(z - z_{hub})\right), \tag{9}$$

where a wind veer of $\alpha_{tt} - \alpha_{bt} > 7°$ across the rotor area provides $\Delta y/D = 0.3$ for the bottom and top tips at $x/D = 5$ (Abkar et al., 2018). The effect of wind veer is not further analysed here, because it has already been studied from field measurements
in Bodini et al. (2017) and Brugger et al. (2019).

## 4   Conclusions

Field measurements of yawed wind turbine wakes with nacelle-mounted scanning Doppler lidars were performed. The wake was characterized in terms of depth, width and deflection from planar and volumetric scans of the Doppler lidars. Together with the inflow measurements, this data was used for validation of the wake deflection and the power predictions of three analytical
models for yawed wind turbine wakes. The observed wake deflection increased with the yaw angle and the comparison to the analytical models showed an overestimation by the Jiménez et al. (2009) model, while the Bastankhah and Porté-Agel (2016) model and the Qian and Ishihara (2018) model matched the measurement data better. The predicted power of the Qian and Ishihara (2018) model had the smallest errors with 17% compared to the SCADA data and 12% compared to the power estimated from the Doppler lidar measurements. Followed by the Bastankhah and Porté-Agel (2016) model with errors of 24%
and 13%, respectively, and the Jiménez et al. (2009) model that had the largest errors with 40% and 28%. For comparison, the power estimated from the Doppler lidar measurements had an error of 14% to the SCADA data. The power coefficient for an inhomogeneous wind field across the rotor area, if the downwind turbine is partially waked, was identified as an error source among others. Further, it was found that some cases of wake steering were detrimental to the power output. The combination of the bias of the wind vane on top of the nacelle when the turbine was yawed, the variability of the wind direction within
the averaging period, and the implemented wake steering design could explain those cases and highlights the importance to develop a wake steering set-up that is robust against those problems. Lastly, it was observed that the wind veer had a dominant effect on the spanwise shape of the wake and kidney-shaped wakes were not observed in the field data. Application of the analytical model to predict the power of waked downstream turbines would benefit from a power coefficient adapted to an inhomogeneous wind field across the rotor area, improved description of the near wake for better handling of short turbine
spacing or low turbulence intensities, and accounting for non-stationary and inhomogeneous boundary layers.

## Appendix A:  Equations of the analytical models

The equations of the three analytical models compared in this article are summarized from their respective publication for convenience.





**Figure 15.** Spanwise cross-sections of the longitudinal velocity field at $x/D = 4$. The top row (a,b) shows velocity deficit from 3D scans of the wake scanning Doppler lidar and bottom row (c,d) shows the results from the model of Zong and Porté-Agel (2020). The left column (a,c) is a case with a positive wind veer of $0.09°$ m$^{-1}$ and the right column (b,d) is a case with a negative wind veer of $-0.06°$ m$^{-1}$.





## A1 Bastankhah and Porté-Agel (2016)

The analytical model from Bastankhah and Porté-Agel (2016) is based on the conservation of momentum and assumes a Gaussian distribution of the velocity deficit. The wake skew angle in the near wake is given by

$$\theta_0 = \frac{0.3\gamma}{\cos(\gamma)}(1 - \sqrt{(1 - C_T \cos(\gamma))}) \tag{A1}$$

with $\gamma$ given in radiant. The length of the near wake is given by

$$x_0 = \frac{\cos(\gamma)(1 + \sqrt{1 - C_T})}{\sqrt{2}(\alpha TI_x + \beta(1 - \sqrt{1 - C_T}))}D \tag{A2}$$

with $\alpha = 2.32$ and $\beta = 0.154$. The width of the wake in the far wake ($x \geq x_0$) is given by

$$\sigma_y(x) = k_y^*(x - x_0) + \frac{\cos(\gamma)}{\sqrt{8}}D \tag{A3}$$

for the vertical direction and by

$$\sigma_z(x) = k_z^*(x - x_0) + \frac{1}{\sqrt{8}}D \tag{A4}$$

for the transversal direction. The wake growth rate is assumed to be isotropic in the spanwise plane and proportional to the 385 turbulence intensity with

$$k_y^* = k_z^* = 0.35TI_x \tag{A5}$$

following results of a field campaign (Fuertes et al., 2018). For $TI_x < 0.06$, the wake growth rates are set to $0.021$ to account for the turbulence induced by the turbine itself. The wake deflection from the line of wind direction at the onset of the far wake is given by

$$\delta_0 = \tan(\theta_0)x_0 \tag{A6}$$

and for the far wake ($x \geq x_0$) by

$$\delta(x) = \delta_0 + \frac{D\tan(\theta_0)}{14.7}\sqrt{\frac{\cos(\gamma)}{k_y^* k_z^* C_T}}(2.9 + 1.3\sqrt{1 - C_T} - C_T)\log\left(\frac{a}{b}\right) \tag{A7}$$

with

$$a = (1.6 + \sqrt{C_T})(1.6\sqrt{\frac{8\sigma_y\sigma_z}{D^2\cos(\gamma)}} - \sqrt{C_T}) \tag{A8}$$

and

$$b = (1.6 - \sqrt{C_T})(1.6\sqrt{\frac{8\sigma_y\sigma_z}{D^2\cos(\gamma)}} + \sqrt{C_T}). \tag{A9}$$

Lastly, the velocity deficit is computed with

$$\frac{\Delta u}{u_{hub}} = \left(1 - \sqrt{1 - \frac{C_T\cos(\gamma)}{8\sigma_y\sigma_z/D^2}}\right)\exp\left(-0.5\frac{(y - \delta)^2}{\sigma_y^2}\right)\exp\left(-0.5\frac{z^2}{\sigma_z^2}\right). \tag{A10}$$





## A2  Qian and Ishihara (2018)

The model of Qian and Ishihara (2018) also uses a Gaussian distribution of the velocity deficit. The different definition of the thrust coefficient used in Qian and Ishihara (2018) is related to definition employed here by $C_T' = C_T \cos(\gamma)$. The wake growth rate is given by

$$k^* = 0.11 C_T'^{1.07} T I_x^{0.20} \tag{A11}$$

and the potential wake width at the rotor plane is given by

$$\epsilon^* = 0.23 C_T'^{-0.25} T I_x^{0.17}. \tag{A12}$$

The wake skew angle in the near wake is given by

$$\theta_{x0} = \frac{0.3\gamma}{\cos(\gamma)} (1 - \sqrt{1 - C_T' \cos^3(\gamma)}) \tag{A13}$$

and the wake width at the onset of the far wake is given by

$$\sigma_{x0} = \sqrt{\frac{C_T'}{\cos(\gamma)} \left( \frac{\sin(\gamma) + 1.88 \cos(\gamma)\theta_{x0}}{44.4\theta_{x0}} \right) D} \tag{A14}$$

with the near wake length given by

$$x_0 = \frac{D}{k^*} \left( \frac{\sigma_{x0}}{D} - \epsilon^* \right). \tag{A15}$$

The wake growth in the far wake is given by

$$\sigma(x) = k^* x + \epsilon^* D \tag{A16}$$

and the wake deflection at the onset of the far wake is given by

$$\delta_{x0} = \theta_{x0} x_0. \tag{A17}$$

The deflection of the wake center from the line of wind direction is given by integration of the wake skew angle in downwind direction (Howland et al., 2016) with

$$\delta(x) = \delta_{x0} + \frac{D\sqrt{C_T'/\cos^2(\gamma)} \sin(\gamma)}{18.24 k^*} \log\left( \frac{c_1}{c_2} \right) \tag{A18}$$

with

$$c_1 = \left( \frac{\sigma_{x0}}{D} + 0.24\sqrt{C_T \cos^3(\gamma)} \right) \left( \frac{\sigma(x)}{D} - 0.24\sqrt{C_T \cos^3(\gamma)} \right) \tag{A19}$$

and

$$c_2 = \left( \frac{\sigma_{x0}}{D} - 0.24\sqrt{C_T \cos^3(\gamma)} \right) \left( \frac{\sigma(x)}{D} + 0.24\sqrt{C_T \cos^3(\gamma)} \right). \tag{A20}$$



The normalized velocity deficit is given by

$$\frac{\Delta u}{u_{hub}} = F(C_T', TI_x, x/D) \exp\left(-\frac{x^2 + (y + \delta(x))^2}{2\sigma^2}\right) \tag{A21}$$

with

$$F(C_T', TI_x, x/D) = (a + bx/D + p)^{-2} \tag{A22}$$

and

$$a = 0.93C_T'^{-0.75}TI_x^{0.17}, b = 0.42C_T'^{0.6}TI_x^{0.2}, p = \frac{0.15C_T'^{-0.25}TI_x^{-0.7}}{(1 + x/D)^2}. \tag{A23}$$

### A3 Jimenez et al. (2009)

The analytical model of Jiménez et al. (2009) is also based on the conversation of momentum, but assumes a top-hat distribution of the longitudinal velocity deficit. The wake growth rate is given by Eq. (A11) and the wake skew angle is given by

$$\theta(x) = \frac{C_T \cos(\gamma)^2 \sin(\gamma)}{2(1 + 2k_w x/D)}. \tag{A24}$$

Integration of the wake skew angle in downwind direction provides the wake deflection, which is given by

$$\delta(x) = \frac{\cos(\gamma)^2 \sin(\gamma) C_T}{4k_w}\left(1 - \frac{1}{1 + 2k_w x/D}\right)D. \tag{A25}$$

The normalized velocity deficit is given by

$$\frac{\Delta u}{u_{hub}} = \frac{C_T D^2 \cos^3\theta}{2(D + k_w x)^2}, \tag{A26}$$

for $\sqrt{(y - \delta)^2 + z^2} \leq D + k_w x$ and zero outside. Other methods to compute the velocity deficit based on a top-hat distribution found in literature were tested, but resulted in larger errors (Peña et al., 2016; Frandsen et al., 2006).

*Author contributions.* Conceptualization: F.P.-A.; Data curation: E.S., D.J., P.B., P.F., and M.D., Formal analysis: P.B.; Funding acquisition: F.P.-A., P.M.; Investigation: A.S., P.B., P.F., and M.M.; Methodology: P.B; Project administration: F.P.-A. and P.M.; Software: P.B, H.Z.; Supervision: F.P.-A., P.M.; Validation: P.B.; Visualization: P.B; Writing - original draft: P.B; Writing - review and editing: P.B., F.P.-A., M.D., A.S.,P.M., and D.J.;

*Competing interests.* The authors declare that they have no conflict of interest.

*Acknowledgements.* Fernando Porté-Agel, Haohua Zong, and Peter Brugger recieved funding from the Swiss National Science Foundation [grant 200021-172538], the Swiss Federal Office of Energy, and the Swiss Innovation and Technology Committee (CTI) within the context



of the Swiss Competence Center for Energy Research FURIES: Future Swiss Electrical Infrastructure. This work was authored in part by the National Renewable Energy Laboratory, operated by Alliance for Sustainable Energy, LLC, for the U.S. Department of Energy (DOE) under Contract No. DE-AC36-08GO28308. Funding provided by the U.S. Department of Energy Office of Energy Efficiency and Renewable Energy Wind Energy Technologies Office. The views expressed in the article do not necessarily represent the views of the DOE or the U.S.

450    Government. The U.S. Government retains and the publisher, by accepting the article for publication, acknowledges that the U.S. Government retains a nonexclusive, paid-up, irrevocable, worldwide license to publish or reproduce the published form of this work, or allow others to do so, for U.S. Government purposes.



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
