# Peer review of "Lidar measurements of yawed wind turbine wakes: characterisation and validation of analytical models"

_Wind Energy Science, 2020_

## Referee Comment (RC1) · Marijn Floris van Dooren (Referee) · 29 May 2020

This paper presents an interesting study on the characterisation and validation of analytical yawed wake models. The writing is of high quality, the figures are nice and in general the paper is very informative. However, in some aspects the paper could be a bit more 'to the point'. I will illustrate that with further comments. I suggest a minor revision.

General comments:

- The conclusions mainly address the errors in the power prediction for different

experimental and analytical methods/models. Even the lowest value of 12% is higher than the expected power improvements reached by wake steering (Sect. 3.3.3). Maybe this could be elaborated a bit more in a broader scope, addressing how these findings contribute to the research field that attempts to increase power production of turbines in an array or wind farm and what are your recommendations on how and with which methods to proceed.

- On the other hand the length of the paper could be reduced a little. I like the fact that the paper is very informative, but sometimes it provides information not directly necessary for the take-away message. One example is Sect. 3.4 on the shape of the wake. Please consider whether it is a vital concern or whether it could be omitted.

- You state that 'studies of yawed wind turbines using field data are rare'. Although this may be true, I recommend you to look into and perhaps cite the work of Bromm (2018), DOI: 10.1002/we.2210 in addition to the other references.

Specific comments:

- P2, L50: What kind of WindCube was used? There are various short-range and long-range WindCube models.

- P11, Fig. 5 and Fig. 6: It would be very good for the overview to see the goodness of fit (correlation) coefficient displayed within the correlations plots.

- P14, Fig. 8: Does the wake center detection function as it should? It seems to jump between the wakes of T2, T3 and T4. Wouldn't it make more sense to try to follow the far wake of T2 instead? Maybe this could be adjusted.

- P17, Fig. 10 and Fig. 11: Again it would be nice to see the correlation coefficient displayed within the figures.

Technical corrections:

- P2, L52: Replace 'thrid' by 'third'.

- P3, L59: Replace 'nacelle' by 'the nacelle'.

- P5, L82: Add 'direction' at the end of the sentence.

- P6, L106: Replace 'StreamLine' with 'Stream Line'.

- P9, L191: Rewrite 'data of either the WindIris or the WindCube was missing'.

- P9, L204: Remove 'the' in front of '07 February 2019'.

- P10, L212: Remove 'the' in front of '11 February 2019'.

- P12, L235: Add 'a' in front of 'mean value'.

- P14, L264: Add 'a' in front of 'correlation coefficient'.

- P16, L268: Replace 'reasons' with 'reason'.

---

## Referee Comment (RC2) · Rebecca Barthelmie (Referee) · 23 Jun 2020

This paper describes a useful set of measurements used to examine wake deflection. Overall it is interesting and has a good message. It could be improved to make it substantially easier to follow and compare the different cases and data sets and models. A data access statement is required by journal policy.

Please see comments below.

Introduction: Could this be quantitative? Rather than listing the papers, wouldn't it be helpful if the introduction gave a background in terms of answering: How big are

power losses due to wakes? What could be expected in terms of the gains from wake steering? What have other modeling studies and the few available field studies indicated are those magnitudes? You could then follow up in the conclusions to evaluate whether a consensus is being reached on the viability of wake steering for power gain for example.

Figure 1. The google map figure needs a scale.

Figure 2 needs an idea at least of how LONG a measurement period this represents. Is it the whole data period i.e. six months of data, from every direction?, every wind speed and turbulence condition ? is it a case study?

Section 2: How was the target yaw offset determined? How were the wind directions determined from the lidar data? What is the purpose of the analytical models? (beyond 'comparison with data'? what is the objective?) Please elaborate why and how you used the models. What are the errors in the wind speed direction comparison? How does that propagate into the uncertainty in the wake deflection analysis?

Section 3.2 How were these wake deflection cases selected? Are you saying it is an analysis of all of the data from January to April?

Please rewrite this section to help readers understand what you mean? What is a favorite in this context?

'First, the wake deflection is verified for non-yawed control cases, where no wake deflection is expected. The distribution of the normalized wake deflection using theWindIris has a RMSE of 0:08 (Fig. 9a) and using the wind direction of theWindCube with the nacelle position of T2 provides a RMSE of 0:07 (Fig. 9b). These errors agree with the RMSE of the yaw angle between 235 the two instruments (4sin(1:30_) = 0:09) and both distributions have mean value that is not significantly different from zero. The consistency between the yaw angle errors and wake deflection distribution shows that the wake scanning and its spatial positioning were working well, and the absence

of a bias shows that the alignment of the wake scanning lidar with the rotor axis is correct (the measured offset of 0:15_ during the installation was taken into account in the processing). Since we could not identify a clear favourite between the WindIris and the WindCube for the yaw angle, the average of both will be used for 240 the remainder of the article.'

Figure 7. Please add some quantitative comparison e.g. correlation coefficients, RMSE? How many measurements are included? Or excluded? How were they selected? It looks like about 30 measured points?

Figure 8. This figure is probably key but again its very difficult to understand. Describe how you chose this case, describe how and where the measurements are located, describe how and where the models were implanted including the derivation of the freestream and its errors. Is tis a totally random case? Was it selected for some specific purpose?

Can you start by laying out the various cases in a table ? Are there are examples, wake steering cases and the complete data set. Are there more? Like the wide case and the narrow case? It is difficult to follow and make comparisons. All of the comparisons should be in a table with the model results to allow a better evaluation? So for example, how does Table 2 compare with Table 1?

In the conclusions please evaluate this study in terms of: 1) Measurement errors vs model errors 2) Magnitude of wake steering vs errors 3) Comparison with other data sets – what is the overall assessment in terms of the viability of wake steering.

Please provide a data access statement.

Please check for typos.

---

## Author Response (AR1)

**Author response to Marijn Floris van Dooren**

**The authors response is shown in red.**

Changes implemented in the new version of the manuscript are in blue.

This paper presents an interesting study on the characterisation and validation of analytical yawed wake models. The writing is of high quality, the figures are nice and in general the paper is very informative. However, in some aspects the paper could be a bit more 'to the point'. I will illustrate that with further comments. I suggest a minor revision.

We thank Dr. van Dooren for his comments, which helped to improve the manuscript. Based on the comments from him and Prof. Barthelmie we implemented the following notable changes to the manuscript:

- The abstract and conclusion were modified to provide more room for the evaluation of the wakesteering setup.
- The validation of the inflow measurements uses the wake-steering cases and the control cases to make the usage of measurement data uniform throughout the results (Sect. 3.1).
- The validation of the analytical model distinguishes various error sources (Sect. 3.3.2)
- The effect of the wake steering on the power is estimated based on the wake-scanning lidar in addition to the analytical model (Sect. 3.4.2).
- Numerous minor changes (additions and clarification within the methods section, spelling and phrasing throughout the manuscript).

The line numbers in our replies refer to the revised manuscript. In addition to the revised manuscript, we also provide a tracked-changes manuscript that visually highlights the changes made.

**General comments:**

• The conclusions mainly address the errors in the power prediction for different experimental and analytical methods/models. Even the lowest value of 12% is higher than the expected power improvements reached by wake steering (Sect.3.3.3). Maybe this could be elaborated a bit more in a broader scope, addressing how these findings contribute to the research field that attempts to increase power production of turbines in an array or wind farm and what are your recommendations on how and with which methods to proceed.

We have two points concerning the issue of the large model errors in comparison to the power increase due to wake steering:

- 1. The effect of wake steering on the power is now also estimated from the wake-scanning Doppler lidar independent from the analytical model (new Fig. 14 and Table 3). These results show the same behavior as the model. Especially, a reduced wake steering success for wind directions outside a narrow sector between 325° and 335° is consistent with results of the analytical model.
- 2. The increase in power due to wake steering results from the wake deflection, which the model reproduces fairly well. On the other hand, the errors of the analytical model includes the shortcomings of the power curve (one third of the error) and missing atmospheric effects (nonstationary conditions, wind veer etc.), which do not affect the findings for the increase in power directly, because it is gained from a comparison of model vs. model.

We believe the manuscript provides a meaning full contribution to the research field by demonstrating that the implemented wake steering was suboptimal and pointing out the causes. Concerning the latter, the found wind direction bias when the wind turbine is yawed is especially important, because it points to the general problem that the standard instrumentation of a wind turbine seems not sufficient to provide the required input for the wake-steering controller with the needed quality. We assume that the bias of the wind vane comes from the flow in close proximity of the nacelle while it is yawed (i.e. it is not an instrument fault, but the flow at instrument location is systematically not aligned with the wind direction of the free stream during yawed operation). We avoided drawing general conclusions about the success of wake steering, because the wake steering setup was not working as intended as mentioned above. Nevertheless, the wake steering increased the power output in some cases, but probably not as much as possible.

Using a narrower wind direction sector, to which a yaw angle offset is applied, only reduced the impact of the suboptimal wake steering, but is not an optimal solution. A correction of the bias might be possible if it only depends on the yaw angle and the wind speed. A proven solution would be a forward facing Doppler lidar to provide the input measurements (that adds other benefits, too).

The effect of wake steering on the power is computed from the wake-scanning lidar (methods on P9, L185-188; results shown in Fig. 14, Table 3, and P21, L354-359). We made changes to the abstract and conclusions to give more room for the wake steering (P24, L376-387).

• On the other hand the length of the paper could be reduced a little. I like the fact that the paper is very informative, but sometimes it provides information not directly necessary for the take-away message. One example is Sect. 3.4 on the shape of the wake. Please consider whether it is a vital concern or whether it could be omitted.

The section was included for two reasons: (i) after looking at position and depth of the wake, its shape would be the next logical property to investigate, and (ii) it provides insights on the dominant effects that are important to be considered in modeling. However, we are aware that the section is only a side note to the findings, and we decided to move the section into an appendix and reference it in the conclusions.

We moved Sect. 3.4 from the main body of the manuscript into Appendix B and referenced in the conclusions (P24, L390-391). The former Sect. 3.3 is now subdivided into a new Sect. 3.3 that only contains the model validation and a new Sect. 3.4 that contains the effect of wake steering on the power.

• You state that 'studies of yawed wind turbines using field data are rare'. Although this may be true, I recommend you to look into and perhaps cite the work of Bromm (2018), DOI: 10.1002/we.2210 in addition to the other references.

Thank you for pointing out this paper. It was extremely relevant in the context of this paper and was an interesting read.

We added Bromm et al. (2018) into the literature review of the introduction (P3, L41-42).

Specific comments:

• P2, L50: What kind of WindCube was used? There are various short-range and long-range WindCube models.

It was a WindCube-V2 profiling lidar (details have been added in Sect. 2.2.1).

Additional information on the Wind Cube added on P4, L76, and L81-82.

• P11, Fig. 5 and Fig. 6: It would be very good for the overview to see the goodness of fit (correlation) coefficient displayed within the correlations plots.

We added quantitative information on errors and goodness of fits to all figures. Further, in response to comments of Reviewer #2, section 3.1 was modified extensively to make the usage of measurement data uniform throughout the paper.

We included the correlation coefficient in the legends of Fig. 5. The new Fig. 6 includes the RMSE and parameters of a fitted Gaussian.

• P14, Fig. 8: Does the wake center detection function as it should? It seems to jump between the wakes of T2, T3 and T4. Wouldn't it make more sense to try to follow the far wake of T2 instead? Maybe this could be adjusted.

Only the solid part of the white line was the successfully detected wake center. Instances where the wake center detection jumps between the wakes were detected and rejected based on the correlation threshold (the rejected parts were shown as the dashed part of the white line in the original figure).

We removed the dashed part of the white line and show only the solid part to avoid confusion (new Fig. 9a).

• P17, Fig. 10 and Fig. 11: Again it would be nice to see the correlation coefficient displayed within the figures.

We included the correlation coefficient and RMSE in the legends of Fig. 10 and Fig. 11.

Technical corrections:

• P2, L52: Replace 'thrid' by 'third'.

The sentence about the third wake-scanning lidar was removed.

- P3, L59: Replace 'nacelle' by 'the nacelle'.
   Inserted "the" (P3, L64).
- P5, L82: Add 'direction' at the end of the sentence.
   Inserted "direction" (P4, L88).
- P6, L106: Replace 'StreamLine' with 'Stream Line'.
   Inserted a space (P6, L112).
- P9, L191: Rewrite 'data of either the WindIris or the WindCube was missing'.
   Corrected (P10, L208-209).
- P9, L204: Remove 'the' in front of '07 February 2019'.

This sentence became obsolete after reworking section 3.1 to use only the wake-steering cases and control cases.

- P10, L212: Remove 'the' in front of '11 February 2019'.
   Removed "the" (P11, L228).
- P12, L235: Add 'a' in front of 'mean value'.
   Inserted "a" (P13, L246).
- P14, L264: Add 'a' in front of 'correlation coefficient'. After adding the correlation coefficient and RMSE to the Fig. 10, we removed this sentence.
- P16, L268: Replace 'reasons' with 'reason' Removed the "s" (P15, L283).

**Author response to Rebecca Barthelmie**

**The authors response is shown in red.**

Changes implemented in the new version of the manuscript are in blue.

This paper describes a useful set of measurements used to examine wake deflection. Overall it is interesting and has a good message. It could be improved to make it substantially easier to follow and compare the different cases and data sets and models. A data access statement is required by journal policy.

We thank Prof. Barthelmie for her comments, which helped to improve the manuscript. Based on the comments from her and Dr. van Dooren we implemented the following changes to the manuscript:

- An overview of the wake-steering cases and the control cases was added the beginning of Sect. 3 (Table1).
- The validation of the inflow measurements uses the wake-steering cases and the control cases to make the usage of measurement data uniform throughout the results (Sect. 3.1).
- The validation of the analytical model distinguishes various error sources (Sect. 3.3.2)
- The effect of the wake steering on the power is estimated based on the wakescanning lidar in addition to the analytical model (Sect. 3.4.2).
- The abstract and conclusion were modified to provide more room for the evaluation of the wake-steering setup.
- Numerous minor changes (additions and clarification within the methods section, spelling and phrasing throughout the manuscript).

The line numbers in our replies refer to the revised manuscript. In addition to the revised manuscript, we also provide a tracked-changes manuscript that visually highlights the changes made.

Please see comments below.

Introduction: Could this be quantitative? Rather than listing the papers, wouldn't it be helpful
if the introduction gave a background in terms of answering: How big are power losses due to
wakes? What could be expected in terms of the gains from wake steering? What have other
modeling studies and the few available field studies indicated are those magnitudes? You
could then follow up in the conclusions to evaluate whether a consensus is being reached on
the viability of wake steering for power gain for example.

The power losses from wake effects depend on turbine spacing, wind direction, atmospheric stability and turbulence levels. A single fully waked wind turbine can produce 40% less power than a wind turbine in the free stream (Simley et al., 2020; Barthelmie et al. 2010). On a wind farm scale, the power losses also depend on the above mentioned variables, but additionally on how deep a wind turbine is behind the leading row of wind turbines (Barthelmie et al, 2010, Porté-Agel et al. 2013).

For a pair of upstream-downstream turbines, the possible power improvement with wake steering given in literature ranges from 3.5% to 11% (Bartl et al. 2016) depending on turbulence levels and turbine distances. Field experiments showed values improvements of 3.5% (Simley et al. 2020) and 4% (Fleming et al. 2019).

We included quantitative information on the wake losses and the power gains with wake steering to the introduction (P1, L17-19 and P2, L29-33).

• Figure 1. The google map figure needs a scale.

A scale was added to the overview map (P3, Fig. 1).

• Figure 2 needs an idea at least of how LONG a measurement period this represents. Is it the whole data period i.e. six months of data, from every direction?, every wind speed and turbulence condition? is it a case study?

Figure 2 uses all available data from 6 January until 9 April 2019, which is consistent with the period used in the results. All wind speeds and turbulence levels are included as long as the wind turbine and the WindCube were operational.

This information was added to the caption of Figure 2 (P5, Fig. 2).

• Section 2: How was the target yaw offset determined? How were the wind directions determined from the lidar data? What is the purpose of the analytical models? (beyond 'comparison with data'? what is the objective?) Please elaborate why and how you used the models. What are the errors in the wind speed direction comparison? How does that propagate into the uncertainty in the wake deflection analysis?

We believe that some of the raised questions resulted from a bad structure of the section. Therefore, we modified Sect. 2.1 to introduce the measurement site only, and the instruments and measurements are then introduced in Sect. 2.2, separately. To answer the above questions here directly:

- The target yaw offset was precomputed before the campaign based on an optimization with the Flow Redirection and Induction in Steady State (FLORIS) software from NREL as described in Fleming et al. (2019).
- The wind directions were determined from the WindCube using the Doppler beam swinging technique (similar to the profiling lidar used in Lundquist et al., 2017).
- The analytical models are compared with the measurements for the purpose of the validation of the models themselves and to evaluate the efficiency of the wake steering. One-to-one comparisons with the wake scanning Doppler lidars are made by computing the analytical models with the input variables measured by the WindCube and WindIris during each wake steering case.
- The errors of the wind speed and the wind direction measurements are analyzed in section 3.1 (which was modified heavily to make it easier to follow and uniform in terms of used measurement data).
- We assume that the RMSE between the WindCube and the WindIris as the error of the yaw angle, which is then propagated to the resulting error of the wake deflection using geometry. The resulting error is shown in Fig. 9b as errorbars.

A reference to the section introducing the instruments and their measurements was added to section 2.1 (P3, L56-57). We added how the target yaw offset function was determined (P3,

L62-63). The purpose of analytical models and how they are computed were added to Sect. 2.4 (P9, L190-191, L193-195, and P10, L203-206).

• Section 3.2 How were these wake deflection cases selected? Are you saying it is an analysis of all of the data from January to April? Please rewrite this section to help readers understand what you mean? What is a favorite in this context?

'First, the wake deflection is verified for non-yawed control cases, where no wake deflection is expected. The distribution of the normalized wake deflection using the WindIris has a RMSE of 0:08 (Fig. 9a) and using the wind direction of the Wind-Cube with the nacelle position of T2 provides a RMSE of 0:07 (Fig. 9b). These errors agree with the RMSE of the yaw angle between 235 the two instruments (4sin(1:30\_) =0:09) and both distributions have mean value that is not significantly different from zero. The consistency between the yaw angle errors and wake deflection distribution shows that the wake scanning and its spatial positioning were working well, and the absence of a bias shows that the alignment of the wake scanning lidar with the rotor axis is correct (the measured offset of 0:15\_ during the installation was taken into account in the processing). Since we could not identify a clear favourite between the WindIris and the WindCube for the yaw angle, the average of both will be used for 240 the remainder of the article.'

Both, the wake-steering cases and the control cases, are selected based on the criteria outlined in section 2.3.1. Briefly summarized:

- The wake-steering cases require a northwestern wind direction with T3 downstream of T2 and active wake steering.
- The control cases require a northeastern wind direction such that T2 will not yaw (limiting to northeaster directions also ensures that the inflow is undisturbed by other wind turbine wakes or topography as for the wake steering cases).

There are 81 wake-steering cases and 76 control cases between 6 January 2019 and 9 April 2019 that fulfill the criteria of section 2.3.1. Their numbers are then further reduced by removing cases with unsuccessful wake center detections (due to insufficient SNR or bad Gaussian fits). A summary is presented in Table 1 below. We reworked Sect. 3.1, to use only the wake-steering cases and the control cases to be consistent with the remainder of the results section (it used all measurement data previously).

With "no clear favorite", we wanted to express that the yaw angles measured by the WindIris and the WindCube compared well with each other and had no biases or any other apparent problem. Therefore, we had no reason to pick one over the other and instead used the average of both.

An overview of the cases has been added at the beginning of the results section (P10, L212-213 and Table 1). Section 3.1 was modified to use data of the wake-steering cases and the control cases to make it consistent with remainder of the manuscript. Fig. 8 was modified to include the 3D scans of the control cases (analog to Fig. 9 for the wake-steering cases). It is specified in each figure caption which data is used (Fig. 5, 6 and 7). Section 3.2 was restructured and rephrased (P13, L243-251).

Table 1: Overview of wake-steering cases and control cases. From top to bottom: the number of 30-minute periods that met the requirements of Sect. 2.3.1, the number of cases with a sufficient SNR of the wake-scanning lidar, the number of cases with a successful detection of the wake center based on the correlation threshold (Sect. 2.3.3), and the number of cases for

which the model prediction of  $u_{mod}$  was possible (Sect. 2.4). The numbers outside of the brackets are the total number of cases, and the numbers inside the brackets are the 2D scans and 3D scans of the wake-scanning lidar, respectively.

|                                               | Wake-steering cases | Control cases |
|-----------------------------------------------|---------------------|---------------|
| Cases based on Sect. 2.3.1                    | 81 (36+45)          | 76 (27+45)    |
| Cases with a sufficient SNR                   | 56 (27+29)          | 66 (26+40)    |
| Cases with a successful wake center detection | 29 (16+13)          | 55 (21+34)    |
| Cases with a prediction of $u_{mod}$          | 41 (19+22)          | -             |

• Figure 7. Please add some quantitative comparison e.g. correlation coefficients, RMSE? How many measurements are included? Or excluded? How were they selected? It looks like about 30 measured points?

Quantitative measures of the comparison have been added and the data is selected according to the criteria outlined in section 2.3.1, which is now stated at the beginning of the results section (see our response to the previous comment). The figure shows 29 data points for the wake-steering cases and 55 for the control cases.

The RMSE and the correlation coefficient have been included to Fig. 7 and the caption states which data is used.

• Figure 8. This figure is probably key but again its very difficult to understand. Describe how you chose this case, describe how and where the measurements are located, describe how and where the models were implanted including the derivation of the freestream and its errors. Is tis a totally random case? Was it selected for some specific purpose?

This case was chosen, because it has the largest yaw offset of the data set and therefore the largest magnitude of the wake deflection, which has two benefits: (i) the errors of the wake deflection are smallest relative to the wake deflection and (ii) the deflection is easy to visually observe. The models were computed from the inflow measurements of the WindCube and WindIris taken at the same time as the wake scanning as described in Sect. 2.4.

We analyzed the influence of the measurement errors on the model error. The analytical models require  $\gamma$ ,  $TI_{WI}$ ,  $u_{WC}(z)$ , and the nacelle position of T2 from the SCADA data as input. In Sect. 3.1., the RMSE for  $\gamma$  and  $u_{WC}$  were determined as 1.42° and 0.42 m/s, respectively. We assume that the nacelle position from the SCADA data is virtually free of errors based on agreement with the position of hard targets in the scan field of the wake-scanning lidar. We could not do a validation for  $TI_{WI}$  and assume an accuracy of 0.015 as given in the manufacturer specifications instead. We quantified the error propagation by varying the model input based on the measurement errors for all investigated wake-steering cases. The errors resulting from  $TI_{WC}$  and  $\gamma$  led to uncertainties of 20 kW and 6 kW, respectively. The error resulting from  $u_{WC}$  resulted in the largest uncertainty with 61 kW.

A detailed description of the example case was added to the manuscript text (P13, L252-260). After the restructuring section 3.2, the example case is now shown in Fig. 9a.

• Can you start by laying out the various cases in a table? Are there are examples, wake steering cases and the complete data set. Are there more? Like the wide case and the narrow case? It

is difficult to follow and make comparisons. All of the comparisons should be in a table with the model results to allow a better evaluation? So for example, how does Table 2 compare with Table 1?

We added a table with an overview of the cases at the beginning of the results section. In addition, we made the usage of the measurement data uniform by modifying Sect. 3.1. Now, the results only use the wake-steering cases and the control cases (and examples chosen from the wake-steering cases).

Only Section 3.4 deviates from this structure as explained at the beginning of its subsections (which includes Fig. 12 and Table 2). In Sect. 3.4.2, we compare a period with wake steering to a period without wake steering, which is not possible based on the definition of the wake-steering cases. In Sect. 3.4.2, we only subdivide the wake-steering cases based on the wind direction to illustrate that the wake steering setup was suboptimal.

An overview of the used measurement data was added at the beginning of the results section (P10, L212-213 and Table 1). Section 3.1 was reworked overall to make usage of the measurement data uniform and easier to follow. The captions of Figures 5, 6, 10, 11, 13, and 14 as well as Table 3 were updated to state which date they are using.

• In the conclusions please evaluate this study in terms of: 1) Measurement errors vs model errors 2) Magnitude of wake steering vs errors 3) Comparison with other data sets – what is the overall assessment in terms of the viability of wake steering.

Regarding the model errors and the magnitude of the wake steering effect:

- The effect of wake steering on the power is now also estimated from the wakescanning Doppler lidar independent from the model (Fig. 14a and Table 3). These results show the same behavior as the analytical model. Especially, a reduced wake steering success for wind directions outside a narrow wind direction range between 325° and 335° is consistent with results of the analytical model.
- The effect of wake steering depends mainly on the deflection of the wake, which the model can predict fairly well. The model errors for the power prediction also includes contributions from the power coefficient and nonstationary conditions, that do not directly enter into the model-to-model comparison from which the wake steering is evaluated. The shortcomings of the power coefficient for a partially waked turbine is responsible for a third of the model error. We reworked Sect. 3.3.2 to better distinguish the various error sources.

In summary, the found model errors do include contributions from the measurement errors, but they do not prevent the evaluation of the wake-steering setup. We believe that the identified problem areas of the model are important conclusions themselves. E.g., a study by Walker et al. (2016) does not list the power coefficient as an important error source in a model validation for wake losses.

Another important conclusion of the paper is that the implemented wake steering was performing suboptimal due to a bias of the wind direction perceived by the wind turbine once it had a yaw offset among other issues. This point is better highlighted in the conclusions, now. We believe that our study is not suited to provide an overall assessment of the viability of wake steering for two reasons: (i) as mentioned above, the investigated wake-steering setup was

not working as intended, and (ii) the investigated wake-steering cases only cover a limited range of atmospheric conditions (e.g., the employed methods limited the analysis to stationary conditions). However, we found that the wake steering improved the power output in some cases despite the issues with the wind vane on top of the nacelle.

This conclusion is in line with other studies. For example Vollmer et al. (2016) already concluded that the wake steering success is sensitive to the wind direction input. Simley (2020) came to similar conclusions based on a SCADA data driven approach, but without the wake-scanning lidar showing the wake position relative to the downstream turbine.

The conclusions were modified to and place a stronger emphasis on the suboptimal wakesteering setup and possible remedies (P24, L376-387). The influence of measurement conditions and measurement errors on the model errors is included to the conclusions (lines P24, L372-375).

• Please provide a data access statement.

A data statement is now included in the manuscript (P24, L394).

• Please check for typos.

A lot of typos were corrected throughout the manuscript based on feedback of a native English speaker. The changes are too numerous to be listed fully.

**Literature**

Barthelmie RJ, Pryor SC, Frandsen ST, Hansen KS, Schepers J, Rados K, Schlez W, Neubert A, Jensen L, Neckelmann S (2010) Quantifying the impact of wind turbine wakes on power output at offshore wind farms. J Atmos OceanTechnol 27(8):1302–1317

Lundquist, J. K., Wilczak, J. M., Ashton, R., Bianco, L., Brewer, W. A., Choukulkar, A., Clifton, A., Debnath, M., Delgado, R., Friedrich, K., Gunter, S., Hamidi, A., Iungo, G. V., Kaushik, A., Kosovi´c, B., Langan, P., Lass, A., Lavin, E., Lee, J. C.-Y., McCaffrey, K. L., Newsom, R. K., Noone, D. C., Oncley, S. P., Quelet, P. T., Sandberg, S. P., Schroeder, J. L., Shaw, W. J., Sparling, L., Martin, C. S., Pe, A. S., Strobach, E., Tay, K., Vanderwende, B. J., Weickmann, A., Wolfe, D., and Worsnop, R.: Assessing State-of-the-Art Capabilities for Probing the Atmospheric Boundary Layer: The XPIA Field Campaign, B. Am. Meteorol. Soc., 98, 289–314, https://doi.org/10.1175/BAMS-D-15-00151.1, https://doi.org/10.1175/BAMS-D-15-00151.1, 2017.

Porté-Agel F, Wu YT, Chen CH (2013) A numerical study of the effects of wind direction on turbine wakes and power losses in a large wind farm. Energies 6(10):5297–5313

Fleming P, King J, Dykes K, Simley E, Roadman J, Scholbrock A, Murphy P, Lundquist J K, Moriarty P, Fleming K, van Dam J, Bay C, Mudafort R, Lopez H, Skopek J, Scott M, Ryan B, Guernsey C and Brake D 2019 Wind Energy Science 4 273-285.

Carbajo Fuertes, F.; Markfort, C.D.; Porté-Agel, F. Wind Turbine Wake Characterization with Nacelle-Mounted Wind Lidars for Analytical Wake Model Validation. Remote Sens. 2018, 10, 668.

**Lidar measurements of yawed wind turbine wakes: characterisation and validation of analytical models**

Peter Brugger1, Mithu Debnath2, Andrew Scholbrock2, Paul Fleming2, Patrick Moriarty2, Eric Simley2, David Jager2, Mark Murphy2, Haohua Zong1, and Fernando Porté-Agel1 1Wind Engineering and Renewable Energy Laboratory (WiRE), École Polytechnique Fedérale de Lausanne (EPFL), 1015 Lausanne, Switzerland 2National Renewable Energy Laboratory (NREL), 15013 Denver West Parkway, Golden, Colorado, 80401, USA **Correspondence:** Peter Brugger (peter.brugger@epfl.ch)

Abstract. Wake measurements of a scanning Doppler lidar mounted on the nacelle of a yawed full-scale wind turbine are during a wake-steering experiment were used for the characterization characterisation of the wake flow, the evaluation of the wake flow, the evaluation of the wake-steering setup, and the validation of analytical wake models. Inflow scanning Doppler lidars, a meteorological mastand the data, and the supervisory control and data acquisition (SCADA) system of the wind turbine control system complemented

- 5 the set-up. Results complemented the setup. Results from the wake-scanning Doppler lidar showed an increase of the wake deflection with the yaw anglethat agreed with two of the three compared models. For yawed cases, the predicted power of a waked downwind turbine estimated by the two previously successful models had an error of 17% and 24% compared to the SCADA data and 12% and 13% compared to the power estimated from the Doppler lidar measurements. Shortcomings of the method to compute the power coefficient in an inhomogeneous wind field are likely the reason for disagreement between estimates using
- 10 the Doppler lidar data versus SCADA data. Further, it was found that some wake steering cases were detrimental to the power output due to errors, and that the wake deflection was not in all cases beneficial for the power output of a downstream turbine due to a bias of the inflow wind direction perceived by the yawed wind turbine and the wake steering design implemented. Lastly, it was observed that the spanwise cross-section Both observations could be reproduced with an analytical model that was initialized with the inflow measurements. Error propagation from the inflow measurements that were used as model input,
- 15 and the power coefficient of a waked wind turbine contributed significantly to the model uncertainty. Lastly, the spanwise cross section of the wake is was strongly affected by wind veer, masking the kidney-shaped wake cross-sections observed from wind-tunnel experiments and numerical simulations effects of the yawed wind turbine on the wake cross sections.

**1 Introduction**

Wind turbines in wind farms can influence other turbines downstream and impact their performance. The interaction of the turbine rotor blades and the wind field creates a spatial volume of reduced wind speed and increased turbulence levels downstream of a wind turbine that can extend for several rotor diameters (Vermeer et al., 2003). This region is called the wake and affects downwind turbines negatively by decreasing power production and increasing fatigue loads (Thomsen and Sørensen, 1999). The spatial proximity of wind turbines in a wind farm, and the wake effects on downwind turbines<del>are an important source</del>, are important sources of power losses (Barthelmie et al., 2010). (Barthelmie et al., 2010). The magnitude of the power loss depends

25 on wind direction, turbine spacing, wind speed, turbulence levels, and atmospheric stability (Stevens and Meneveau, 2017). In case of a fully waked wind turbine losses around 40% compared to a wind turbine in the free flow have been observed (Barthelmie et al., 2010; Simley et al., 2020).

Mitigating these wake effects on downwind turbines is an ongoing focus of research. Strategies that have been proposed are adjusting the blade pitch angle and the generator torque (Bitar and Seiler, 2013), counter-rotating rows of wind turbines

- 30 in wind farms (Vasel-Be-Hagh and Archer, 2017), optimizing the placement of the turbines within the wind farm based on terrain and wind climate (e.g. Shakoor et al., 2016; Kuo et al., 2016) (e.g., Shakoor et al., 2016; Kuo et al., 2016), or deflecting the wake away from the downwind turbine by introducing a yaw offset to the upwind turbine (Medici and Dahlberg, 2003). The latter approach, called wake steering or active yaw control, is the focus of this paper. It utilizes the thrust force that the rotor imposes on the flow and, by offsetting the rotor from the flow direction, a transverse component of the thrust force is generated
- that displaces the wake from the line of the wind direction with the goal to deflect it away from the downwind turbine. While the power production of the yawed turbine is reduced, this loss is potentially overcompensated for by the power gains of the downwind turbine (Bastankhah and Porté-Agel, 2015), and the strategy can be extended to a full wind farm (Gebraad et al., 2016). Wind-tunnel studies of wake steering showed an increase in power for the combined upstream-downstream turbine pair between 3.5% and 11% depending on inflow turbulence level and turbine separation distance (Bartl et al., 2018) and a field test
  at two commencies are an extended on increase by 4% (Elemine et al., 2010).

40 at two commercial wind turbines showed an increase by 4% (Fleming et al., 2019).

Analytical models describe the wake of a yawed wind turbine based on a set of turbine and inflow parameters (Jiménez et al., 2009; Bastankhah and Porté-Agel, 2016; Qian and Ishihara, 2018). These models are computationally cheap compared to numerical simulations and therefore can be used to find a set of yaw angles that maximizes the power output (Gebraad et al., 2016; Fleming et al., 2019). Validation of the analytical models for yawed wind turbine wakes and studies on

45 the effectiveness of the wake steering have been done with wind tunnel experiments (e.g. Bastankhah and Porté-Agel, 2016) (e.g., Bastankhah and Porté-Agel, 2016) and numerical simulations (e.g. Vollmer et al., 2016)(e.g., Vollmer et al., 2016). However, studies of yawed wind turbines using field data are rare: Fleming et al. (2017a) and Annoni et al. (2018) analysed the wake deflection, the wake recovery, and the power output for an isolated yawed turbine; Fleming et al. (2017b) investigated the effects of wake steering on the power production for a yawed upwind and a waked downwind turbines at an offshore-site;

50 and turbine at a land-based site; Bromm et al. (2018) investigated the wake deflection of a yawed turbine with remote-sensing instruments with detailed error analysis; most recently Simley et al. (2020) investigated the influence of the wind direction variability on the achieved yaw offsets and power gains based on the supervisory control and data acquisition (SCADA) data.

In this paper, field measurements, including inflow and wake measurements as well as wind turbine control system SCADA data from a wake steering wake-steering upwind turbine and a waked downwind turbine, are used to: (i) characterize characterize

[revised manuscript text omitted]